# Investigating the dependence of mineral dust depolarization on complex refractive index and size with a laboratory polarimeter at 180.0° lidar backscattering angle

Alain Miffre[1], Danaël Cholleton[1], Clément Noël[1] and Patrick Rairoux[1]

[1]University of Lyon, Université Claude Bernard Lyon 1, CNRS, Institut Lumière Matière, F-69622, Villeurbanne, France

*Correspondence to*: Alain Miffre (alain.miffre@univ-lyon1.fr)

**Abstract.** In this paper, the dependence of the particles depolarization ratio ($PDR$) of mineral dust on the complex refractive index and size is for the first time investigated through a laboratory $\pi$-polarimeter operating at 180.0° backscattering angle and at (355, 532) nm wavelengths for lidar purposes. The dust $PDR$ is indeed an important input parameter in polarization lidar experiments involving mineral dust. Our $\pi$-polarimeter provides sixteen accurate (< 1 %) values of the dust lidar $PDR$ at 180.0° corresponding to four different complex refractive indices, studied at two size distributions (fine, coarse) ranging from 10 nm to more than 10 µm, and at (355, 532) nm wavelengths, while accounting for the highly irregular shape of mineral dust, which is difficult to model numerically. At 355 nm, the lidar $PDR$ of coarser silica, the main oxide in mineral dust, is equal to $(33 \pm 1)$ % while that of coarser hematite, the main light absorbent in mineral dust, is $(10 \pm 1)$ %. This huge difference is here explained by accounting for the high imaginary part of the hematite complex refractive index. In turn, Arizona dust exhibits higher depolarization than Asian dust, due to the higher proportion in hematite in the latter. As a result, when the strong light absorbent hematite is involved, the dust lidar $PDR$ primarily depends on the particles complex refractive index and its variations with size and shape are less pronounced. When hematite is less or not involved, the dust lidar $PDR$ increases with increasing sizes, though the shape dependence may then also play a role. The (355, 532) nm wavelength dependence of the dust lidar $PDR$ then allows discussing on the involved particle sizes, thus highlighting the importance of dual-wavelength (or more) polarization lidar instruments. We believe these laboratory findings will help improving our understanding of the challenging dependence of the dust lidar $PDR$ with complex refractive index and size to help interpret the complexity and the wealth of polarization lidar signals.

## 1 Introduction

With worldwide annual emissions between 1000 to 3000 Tg (Monge et al., 2012), mineral dust is a highly important constituent of the atmosphere, which contributes to ice cloud formation by acting as a freezing nucleus and to the carbon cycle by fertilizing nutrient poor ecosystems such as the Amazon rainforest after long-range transport (Bristow et al., 2010). As underscored in the latest IPCC report (2021), mineral dust also contributes to the Earth's radiative budget through light scattering and

absorption, by reducing the amount of energy reaching the Earth's surface (Kosmopoulos et al., 2017). The radiative impact
associated with a Saharan dust storm has been recently quantified by Francis et al. (2022). This climatic impact is however
subject to large uncertainties, mainly due to the great complexity in size, shape and mineralogy of mineral dust. In the
atmosphere, the size distribution of mineral dust is mainly determined by the distance from the dust source region. Two freshly
uplifted dust aerosols may indeed exhibit different size distributions at far-range remote sites (Ryder et al., 2013), due to the
rapid removal of the largest particles by gravitational settling. Mineral dust particles also exhibit a high degree of complexity
in shape. Electron microscopic images (Kandler et al., 2011) indeed highlight the nonspherical and highly irregular shape of
mineral dust particles, with sharp edges, sometimes even surface roughness (Nousiainen, 2009). The mineral dust surface is
itself subject to photo-catalytic reactions leading even to new particle formation events (Dupart et al., 2012). The third degree
of complexity of mineral dust related to this study lies in its mineralogy. Mineral dust indeed consists in a heterogeneous
mixture of various chemical oxides among which the most predominant is silica oxide. Aluminum and iron oxides are also
present in proportions depending on the dust source region. As an example, the desert in Central Australia is iron oxides rich
(Bullard and White, 2002). This diverse mineralogy results in a diversity of complex refractive indices for mineral dust.

In the atmosphere, mineral dust is additionally often mixed with other aerosols. To face such a complexity, ground and satellite-
based polarization lidar instruments, based on light backscattering by nonspherical particles, have been developed
(Freudenthaler et al., 2009; Tesche et al., 2009; Sugimoto and Lee, 2006; Winker et al., 2009; Miffre et al., 2019; Hofer et al.,
2020; Hu et al., 2020) to discern the mineral dust contribution to two-component particles external mixtures, by applying lidar
partitioning algorithms such as the $1\beta + 1\delta$ algorithm (Tesche et al., 2009; Mehri et al., 2018). Such lidar-based retrievals are
however under-constrained and depend on prior knowledge regarding input parameters such as the lidar particles'
depolarization ratio ($PDR$). The lidar $PDR$ quantifies the mineral dust particles deviation from isotropy and is key for aerosol
typing (Hofer et al., 2020; Burton et al., 2012). As explained in light scattering textbooks (Bohren and Huffman, 1983;
Mishchenko et al., 2002), it depends on the particles size, shape and complex refractive index. The size dependence of the
lidar $PDR$ was studied in field by Hofer et al. (2020). The downside of such field measurements is that the observed aerosol is
nevertheless that of a particles mixture, which may induce some discrepancies in the retrieved dust lidar $PDR$ (Miffre et al.,
2011). As an alternate, for accurate retrievals of the mineral dust lidar $PDR$, light backscattering numerical simulations have
been developed, by assuming a particles shape model such as the spheroidal shape model, computed with the T-matrix
numerical code (Mishchenko and Travis, 1998), as successfully applied for mineral dust during the SAMUM field campaign
(Müller et al., 2013) or, by considering more realistic shapes, based on stereograms, computed with the discrete-dipole-
approximation (Lindqvist et al., 2014; Gasteiger et al., 2011). Depending on the assumed shape model, the lidar $PDR$ can be
very different with induced variations in the lidar-retrieved dust mass concentrations (Mehri et al., 2018). Recently, Luo et al.
(2022) and Huang et al. (2022) discussed on the ability of the spheroidal model to mimic the complex shape of mineral dust.
Likewise, Zubko et al. (2013) found spheroids inadequate for describing the dust particles' spectral dependence of the lidar
$PDR$. Such light scattering numerical simulations nonetheless rely on simplifying assumptions that should be carefully
checked. Laboratory experiments on natural dust samples at 180.0° lidar exact backscattering angle are then looked-for as they
provide quantitative evaluations of the mineral dust lidar *PDR* within experimental error bars. Indeed, in laboratory, the
retrieved lidar *PDR* is, by construction, that of pure mineral dust and the dependence of the dust lidar *PDR* with size and
mineralogy can be evaluated. Moreover, the complex shape of mineral dust is then accounted for. However, existing laboratory
light scattering experimental set-ups (Glen and Brooks, 2013; Järvinen et al., 2016; Gautam et al., 2020; Liu et al., 2020;
Kahnert et al., 2020; Gómez Martín et al., 2021) can only provide approximate values of the dust lidar *PDR* for the following
reasons:
–  Such apparatuses operate at near backscattering angles only (< 180.0°), without covering the exact lidar
backscattering angle of 180.0°. The retrieved lidar *PDR* is then extrapolated to 180.0° following simplifying
numerical assumptions, ignoring the complexity in shape of mineral dust (Liu et al., 2020; Gómez Martín et al., 2021).
To provide accurate values of the dust lidar *PDR*, such assumptions must be carefully discussed as  the lidar *PDR*
actually depends on the scattering angle in an unpredictable way, as underscored in light scattering textbooks (Bohren
and Huffman, 1983; Mishchenko et al., 2002), due to the complex shape of mineral dust. For that, a laboratory
measurement of the dust lidar *PDR* at 180.0° is mandatory.
–  Also, most of the above apparatuses operate at a single wavelength, either 442, 488, 552, 632, 647 or 680 nm, which
differs from the (355, 532, 1064 nm) wavelengths which are applied in polarization lidar field experiments. As for
Raman lidars, such wavelength extrapolations up to the (355, 532, 1064 nm) lidar wavelengths are a source of
discrepancy as the dust lidar *PDR* actually depends on the complex refractive index, which is wavelength dependent
(Bohren and Huffman, 1983; Mishchenko et al., 2002). For that, a laboratory measurement at the lidar wavelengths
is mandatory.
In this paper, accurate values (< 1%) of the dust lidar *PDR* are provided from a laboratory π-polarimeter operating at 180.0°
lidar exact backscattering angle and at 355, 532 nm wavelength, to account for the importance of the spectral dependence of
the lidar *PDR* to better constrain lidar inversions and aerosol typing (Burton et al., 2016; Haarig et al., 2022). Since the
scattering angle and the wavelengths are determined for lidar purposes, we here investigate the dependence of the mineral dust
lidar *PDR* on the dust particles size and complex refractive index (*CRI*), the latter being particularly important as related to
light absorption. Light absorption by mineral dust preferentially occurs in the UV and VIS spectral domains, being nearly null
in the near-infrared spectral range (Di Biagio et al., 2019), noticeably in the presence of iron oxides (Formenti et al., 2014;
Caponi et al., 2017). By absorbing short-wave radiations, such oxides hence play a critical role in determining the overall
impact of dust aerosol on climate forcing (Go et al., 2022). We hence focused on 355 and 532 nm lidar wavelengths and
considered four dust samples differing in their *CRI*, thus in mineralogy: i) silica oxide ($SiO_2$), as the most abundant mineral
oxide present in mineral dust, ii) iron oxide (hematite, $Fe_2O_3$), as the main light absorbent present in mineral dust (Gautam et
al., 2020; Zong et al., 2021; Go et al., 2022), iii) and iv) two heterogeneous mixtures of the above two oxides in various
proportions, as detailed in Section 2. The dependence of the lidar $PDR$ with size is then likewise investigating by accounting
for the fine and coarse modes of the particles size distribution ($SD$), to which lidar instruments are sensitive (Mamouri and
Ansmann, 2017), thus extending the size range of our previous laboratory findings (Miffre et al., 2016) to particles sizes larger
than 800 nm and to other mineralogy, as asked for in Tesche et al. (2019). According to the manufacturer, the size distribution
of our dust samples ranged from 10 nm to more than 10 μm in diameter. Our work provides sixteen laboratory-derived accurate
dust lidar $PDR$ values, corresponding to four mineral dust samples differing in mineralogy, given at two $SD$ (fine, coarse) and
at two wavelengths (355, 532 nm).  Moreover, the role of the imaginary part of the hematite $CRI$, which may lead to
modifications in the lidar $PDR$, is here for the first time quantified and discussed.

The paper is structured as follows. In Section 2, the complex refractive indices and size distributions of our four dust samples
are presented. The laboratory $\pi$-polarimeter at 180.0° lidar backscattering angle is then presented in Section 3, together with
the dust lidar $PDR$ retrieval methodology, derived from the scattering matrix formalism (Mishchenko et al., 2002). The main
findings are outlined in Section 4 where the sixteen values of dust lidar $PDR$ are given and a discussion is proposed to
investigate the dependence of the dust lidar $PDR$ on the imaginary part of the dust $CRI$. As in elastic lidar applications, we
here consider the elastic backscattering of electromagnetic radiation of wavelength $\lambda$ by an ensemble of mineral dust particles
of complex refractive index $m = n + i\kappa$ embedded in ambient air.
**2. Mineral dust samples**
**2.1 Refractive indices**
Mineral dust is a complex mixture of several chemical oxides presenting various complex refractive indices. To investigate
the dependence of the dust lidar $PDR$ on the complex refractive index  ($CRI$), we consider the four following case studies :

–  Silica, or silicon oxide ($SiO_2$) is here considered as being the main pure chemical component present in mineral dust.
The silica $CRI$ as given by Longtin et al. (1988) is equal to 1.546, hence exhibiting no absorptive component.
–  Iron oxide, or hematite ($Fe_2O_3$), is in contrast here selected as being a climatically significant light absorbent in the
shortwave spectral region, that can be transported far from source regions with similar efficiency as black carbon
particles (Lamb et al., 2021). It recently regained in interest with papers specifically dedicated to this constituent
(Gautam et al., 2020; Zong et al., 2021). Hematite is unique among all chemical oxides present in mineral dust due
its strong $CRI$. Both $n$ and $\kappa$ are large for hematite, with $\kappa$-values more than 100 times those of other soil mineral
components at lidar wavelengths. Hence, hematite dominates absorption while other minerals can be considered as
non-absorbing (Go et al., 2022). The real and imaginary part of the hematite CRI is provided by Scanza et al. (2015):
m = 2.13 + 0.94i at 355 nm wavelength (3.07 + 0.55i at 532 nm wavelength)..
−    Arizona Test Dust (hereafter called Arizona dust) is likewise considered as an example of natural mineral dust sample
that is a mixture of the above two oxides. According to the manufacturer (Power Technology Inc.), Arizona Test Dust
is composed of silica (68-76 %), while hematite is only weakly present in Arizona dust (2-5 %). In short, Arizona
dust is hence rather silica-rich. As given by the manufacturer, the Arizona dust $CRI$ is $m = 1.51 + 10^{-3}i$, without
however any given on its spectral dependency. Effective medium theories can alternately be applied to account for
the sample inhomogeneity as calculated in Miffre et al. (2016), who arrived to $m = 1.57 + 10^{-2}i$ at 355 nm
wavelength and $1.55 + 5.10^{-3}i$ at 532 nm wavelength. As a result, the Arizona dust sample $CRI$ is characterized by
$n \sim 1.5$ and a low absorbing component $\kappa \sim 5.10^{-3}$.
−    Asian dust is finally also considered as an important case study of natural mineral dust sample, presenting however a
lower proportion of silica (34-40 %) and a higher proportion in hematite (17-23 %). For Asian dust, we use a
commercial sample provided by Powder Technology (commercial name: Kanto Loam), commonly used as a dust
interferon in pollen light scattering measurements in Japan (Iwai, 2013), hence representative of observed atmospheric
Asian dust. In this way, we symmetrized our approach by dealing with both Arizona Test Dust and Asian Test Dust.
The CRI of Asian dust, evaluated from effective medium approximation, is m = 1.70 + 0.09i at 355 nm wavelength
and 1.72 + 0.03i at 532 nm wavelength. Hence, compared with Arizona dust, Asian dust is more hematite-rich and
hence exhibits a larger imaginary part for its $CRI$.

Other chemical oxides are also present in our dust samples in various percentages, but with negligible imaginary parts of CRI
compared with that of hematite. Investigating the PDR of these oxides is then beyond the scope of this paper. Their percentage
in (Arizona Test Dust, Asian Dust) is given for clarity: $Al_2O_3$ (11 %, 29 %), CaO (4 %, 1.5 %), $K_2O$ (3.5 %, 0 %), $Na_2O$ (2 %,
0 %), MgO (1.5 %, 5 %), $TiO_2$ (0.5 %, 2 %). The solid dust samples, provided by Sigma Aldrich and Powder Technology
manufacturers, were embedded in laboratory ambient air by using a solid dust generator supplied with dried compressed air
($RH$ < 10 %) to get dry solid dust particles embedded in laboratory ambient air at a constant number concentration, before
injecting the dust samples into the light scattering volume, as presented in Section 3.
**2.2 Size distribution ($SD$)**
For each above dust sample, we consider two size distributions ($SD$) to likewise investigate the dependence of the dust lidar
$PDR$ on the particles size:
−    The coarser $SD$, represented in grey in Fig. 1. This SD is more representative of mineral dust particles close to dust
regions, although it does not cover the full range of large dust particles measured close to dust sources, showing
particles with diameters > 50 μm (Ryder et al., 2019),
–      A finer *SD*, plotted with a black line in Fig. 1, aimed at being more representative of mineral dust particles after long-
160            range transport, i.e. farther from the dust source regions.

The *SD* were obtained by adding / removing a cyclone to our experimental set-up allowing to add / remove particles with
diameter above 800 nm, thus exploring particles size ranges below and above 800 nm, as asked for in Tesche et al. (2019).
More precisely, the two considered SD correspond to a size distribution with and without coarse mode. The *SD* were measured
with an optical particles sizer (OPS 3330) coupled with a scanning mobility particles sizer (SMPS 3081), which selects the
dust particles as a function of their electric mobility, this latter quantity being diameter-dependent. As in Järvinen et al. (2016),
our size instruments could not measure dust particles with diameter above 10 µm. According to the manufacturer, such giant
particles (Ryder et al., 2019) are however present in our dust samples, at a low number concentration. The measured SD are
representative of what is observed in atmosphere, with a low number concentration of more than 10 µm particles, as observed
by Weinzierl et al. (2017). The particles *SD* displayed in Fig. 1 are in agreement with the specifications provided by the
manufacturers.

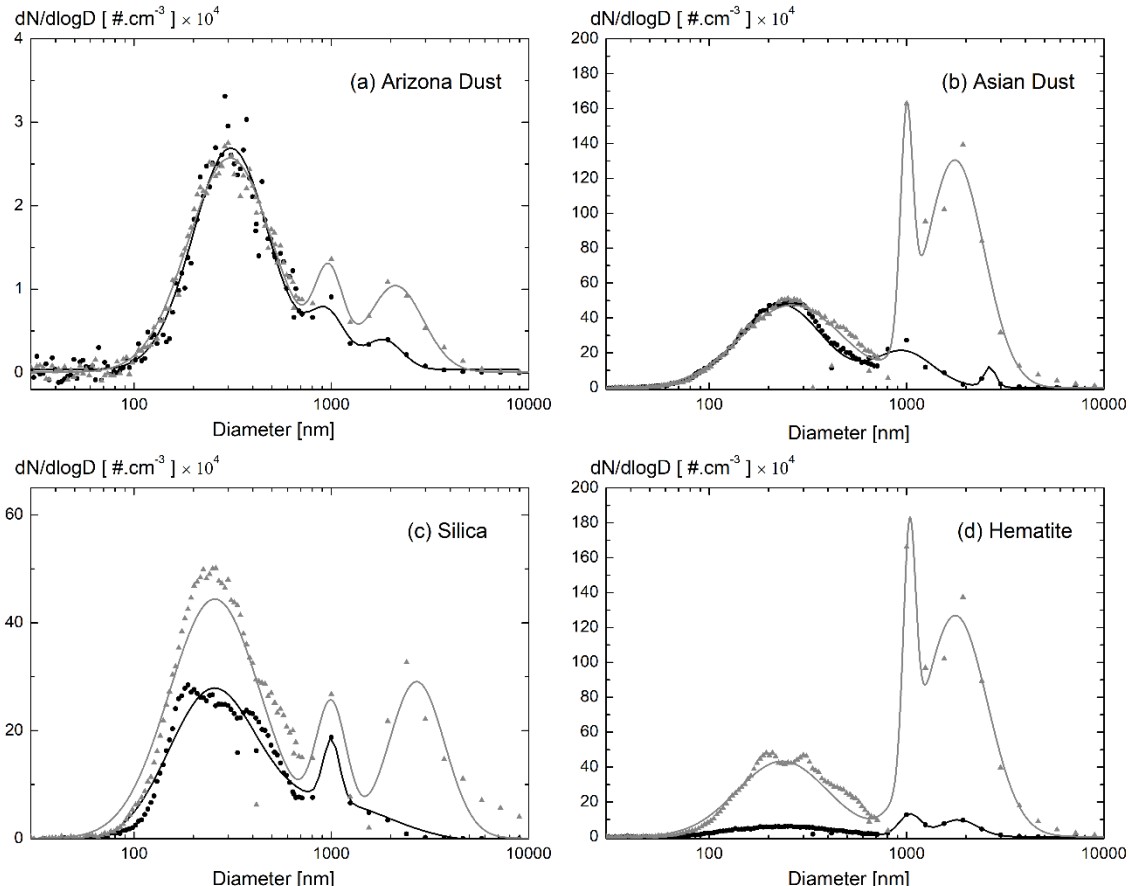


Figure 1: Dust particles size distributions (*SD*) for: (a) Arizona dust, (b) Asian dust, (c) Silica ($SiO_2$), (d) Hematite ($Fe_2O_3$) in
the presence / absence of the added cyclone (finer *SD*, in solid black) / (coarser *SD*, in dotted grey). The retrieved *SD*, obtained
by log-normal adjustments, agree with the specifications provided by the manufacturers.

## 3 Methodology

In this section, we detail our methodology for accurate laboratory evaluations of the dust lidar $PDR$ at lidar exact backscattering angle of 180.0° for accurate lidar $PDR$-retrievals.

### 3.1 Scattering matrix formalism

The dust lidar $PDR$ can be evaluated in the framework of the scattering matrix formalism (Mishchenko et al., 2002; Bohren and Huffman, 1983). In this formalism, the polarization state of the incident and scattered radiations are described by their respective Stokes vectors $\boldsymbol{St_i} = [I_i, Q_i, U_i, V_i]^{\mathrm{T}}$ and $\boldsymbol{St} = [I, Q, U, V]^{\mathrm{T}}$, defined with respect to the scattering plane, used as a reference plane (Mishchenko et al., 2002). The first Stokes component $I$ corresponds to the light intensity, Q and U describe linear polarization, while V accounts for circular polarization. At a distance $d$ from the mineral dust samples, if single-scattering and particles random orientation are assumed, for macroscopically isotropic and mirror-symmetric mediums, the incident and scattered Stokes vectors relate with a bloc-diagonal scattering matrix (Mishchenko et al., 2002; Bohren and Huffman, 1983):

$$\begin{pmatrix} I \\ Q \\ U \\ V \end{pmatrix} = \frac{1}{k^2 d^2} \begin{bmatrix} F_{11,\lambda}(\theta) & F_{12,\lambda}(\theta) & 0 & 0 \\ F_{12,\lambda}(\theta) & F_{22,\lambda}(\theta) & 0 & 0 \\ 0 & 0 & F_{33,\lambda}(\theta) & F_{34,\lambda}(\theta) \\ 0 & 0 & -F_{34,\lambda}(\theta) & F_{44,\lambda}(\theta) \end{bmatrix} \begin{pmatrix} I_i \\ Q_i \\ U_i \\ V_i \end{pmatrix} \tag{1}$$

Where the matrix elements $F_{ij,\lambda}(\theta)$ $(i, j = 1 - 4)$ depend on the wavelength $\lambda$ of the radiation (hereafter noted as a subscript) and carry information on the mineral dust particles size, shape and $CRI$. The scattering angle is $\theta = (\boldsymbol{k_i}, \boldsymbol{k})$, where $k = k_i = 2\pi/\lambda$ is the wave vector of the electromagnetic wave. In lidar applications, the scattering angle is equal to $\pi$ (i.e. exact backscattering angle). To highlight the need for laboratory measurements at the specific 180.0° lidar backscattering angle, near backscattering angles (i.e. $\theta < \pi$) are also considered in this section. Indeed, at $\theta = \pi$, $F_{33,\lambda} = -F_{22,\lambda}$ and $F_{12,\lambda} = F_{34,\lambda} = 0$ (Zubko et al., 2013; David et al., 2013) while $F_{44,\lambda} = F_{11,\lambda} - 2F_{22,\lambda}$ due to the backscattering theorem (van de Hulst, 1957), so that Eq. (1) simplifies as follows for lidar applications:

$$\begin{pmatrix} I \\ Q \\ U \\ V \end{pmatrix} = \frac{1}{k^2 d^2} \begin{bmatrix} F_{11,\lambda}(\pi) & 0 & 0 & 0 \\ 0 & F_{22,\lambda}(\pi) & 0 & 0 \\ 0 & 0 & -F_{22,\lambda}(\pi) & 0 \\ 0 & 0 & 0 & F_{11,\lambda}(\pi) - 2F_{22,\lambda}(\pi) \end{bmatrix} \begin{pmatrix} I_i \\ Q_i \\ U_i \\ V_i \end{pmatrix} \tag{2}$$

As a result, it is only at elastic lidar exact backscattering angle ($\theta = \pi$) that $F_{12,\lambda} = 0$ so that the scattering matrix reduces to only two non-vanishing elements $F_{11,\lambda}(\pi)$ and $F_{22,\lambda}(\pi)$.

## 3.2 Lidar particles depolarization ratio *PDR*

The expression of the so-called particles linear depolarization ratio (*PDR*) at wavelength $\lambda$ and scattering angle $\theta$ can be found in light scattering textbooks (Mishchenko et al., 2002; Schnaiter et al., 2012):

$$PDR_\lambda(\theta) = \frac{1 - F_{22,\lambda}(\theta)/F_{11,\lambda}(\theta)}{1 \pm 2F_{12,\lambda}(\theta)/F_{11,\lambda}(\theta) + F_{22,\lambda}(\theta)/F_{11,\lambda}(\theta)} \tag{3}$$

where the positive (resp. negative) sign corresponds to *p*-polarized (resp. *s*-polarized) incident electromagnetic radiation. The PDR stated in Eq. (3) is the linear PDR, which can be related to the circular PDR if need be (Mishchenko et al., 2002). Since $F_{11,\lambda}$, $F_{12,\lambda}$ and $F_{22,\lambda}$ may vary with the scattering angle, depending on the dust sample, the dust *PDR* at near backscattering angles ($\theta < \pi$) differs from that obtained at specific lidar backscattering angle ($\theta = \pi$). The deviation of $F_{11,\lambda}$, $F_{12,\lambda}$ and $F_{22,\lambda}$ from their value at exact backscattering angle cannot be quantified since no analytical light scattering theory exists for such complex-shaped particles as mineral dust. Therefore, a laboratory experiment at specific lidar exact backscattering angle ($\theta = \pi$) is required for precise evaluations of the dust lidar *PDR*. At specific lidar backscattering angle of $\pi$, Eq. (3) becomes:

$$PDR_\lambda(\pi) = \frac{1 - F_{22,\lambda}(\pi)/F_{11,\lambda}(\pi)}{1 + F_{22,\lambda}(\pi)/F_{11,\lambda}(\pi)} \tag{4}$$

Hence, accurate evaluations of the dust lidar *PDR* rely on accurate determinations of the ratio $F_{22,\lambda}/F_{11,\lambda}$ at specific lidar $\pi-$ angle. As for the ratio $F_{22,\lambda}/F_{11,\lambda}$, the dust lidar *PDR* is size, shape and refractive index dependent and this dependency is discussed in Section 4. Spherical particles, for which $F_{22,\lambda}/F_{11,\lambda} = 1$, lead to $PDR_\lambda(\pi) = 0$. In what follows, to ease the reading, the dust lidar *PDR* will be noted $PDR_\lambda$ without reference to scattering angle ($\theta = \pi$).

## 3.3 Laboratory $\pi$-polarimeter for retrieving the lidar *PDR* of mineral dust

In (Miffre et al., 2016), for the first time to our knowledge, a laboratory $\pi$-polarimeter was built to address light backscattering by aerosol particles. We here recall its main characteristics for clarity. The aerosols $\pi$-polarimeter is schemed in Fig. 2. As in lidar applications, pulsed laser light is used to measure the time-of-flight taken by a laser pulse to reach the dust sample and be detected after light backscattering. The backscattering geometry is set by inserting a well-characterized polarizing beam splitter cube (*PBC*) between the emission and the dust samples, with a precision of 1 mm out of 10 meters to ensure the $\pi$-polarimeter covers the lidar exact backscattering direction with accuracy: $\theta = (180.0 \pm 0.2)°$. The laboratory aerosol $\pi$-polarimeter is actually composed of two identical polarimeters, one per wavelength, to evaluate the lidar PDR of a given dust sample at 355 and 532 nm wavelength simultaneously. Moreover, to decrease the retrieval uncertainty on the dust PDR, the polarization state of the backscattered radiation is analysed for a set of incident polarization states of the incident

light using a quarter-wave plate ($QWP$). To validate the laboratory $\pi$-polarimeter, we carefully checked that homogeneous
spherical particles, such as ammonium sulfate particles, which scatter light as described by the Mie theory (Bohren and
Huffmann, 1983), were indeed providing zero lidar $PDR$ when following the methodology described in the below section.

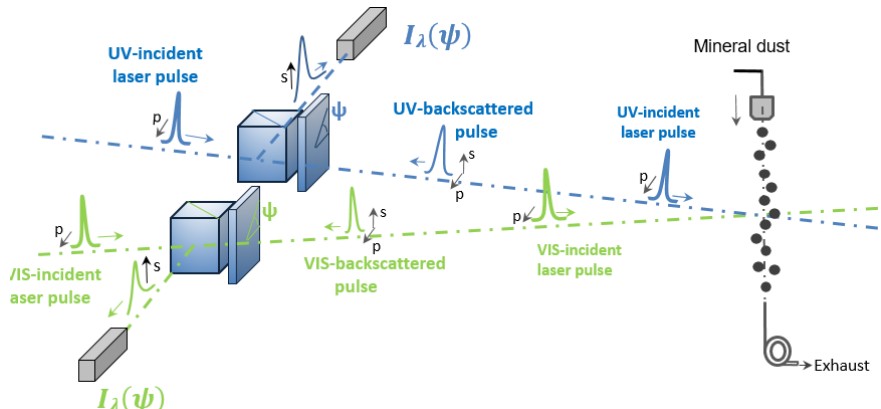


Figure 2: Scheme of the laboratory $\pi$-polarimeter operating at lidar exact backscattering angle of $(180.0 \pm 0.2)°$ allowing accurate retrievals
of the lidar $PDR$ at 355 and 532 nm wavelength simultaneously for an aerosol sample (Miffre et al., 2016). The $(p, s)$ polarization
components are defined with respect to the laser scattering plane and $\psi$ is the angle between the fast axis of the $QWP$ and the laser scattering
plane, counted counter-clockwise for an observer looking from the $PBC$ to the particles. The dust lidar $PDR$ is then evaluated from the ratio
$F_{22,\lambda}/F_{11,\lambda}$ at specific $\pi$-angle, following the methodology described in Section 3.4.

## 3.4 Laboratory retrievals of mineral dust lidar $PDR$

We can formulate the PDR measurements of dust particles, using successive Mueller matrices denoting to the optical
elements of the $\pi$-polarimeter and the scattering medium, encountered by the laser pulse from the laser source to the dust
particles sample then back to the light detector. The measured backscattered intensity is :

$\quad I_\lambda(\psi) = \frac{\eta_\lambda P_\lambda}{d^2}[1,0,0,0]^T[\boldsymbol{PBC}][\boldsymbol{QWP}(-\psi)][\boldsymbol{F_\lambda}][\boldsymbol{QWP}(\psi)][\boldsymbol{PBC}](St_i)$ (5)

Where $\eta_\lambda$ is the optoelectronics efficiency of the light detector and $P_\lambda$ is the laser power density, while $(St_i) = [1,1,0,0]^T$ is
the Stokes vector of the incident laser light. The expression of the dust backscattering matrix $[\boldsymbol{F_\lambda}]$ at wavelength $\lambda$ is is given
in Eq. (2), while $[\boldsymbol{PBC}]$ and $[\boldsymbol{QWP}(\pm\psi)]$ are the Mueller matrices of the $PBC$ and the $QWP$ respectively (Shurcliff, 1962).
To develop Eq. (5), it is then advised to first calculate the raw vector $[1,0,0,0]^T[\boldsymbol{PBC}][\boldsymbol{QWP}(-\psi)][\boldsymbol{F_\lambda}]$ then multiply it with
the Stokes vector of the incident laser light $[\boldsymbol{QWP}(\psi)][\boldsymbol{PBC}](St_i)$ equal to $[1, \cos^2(2\psi), -\sin(4\psi)/2, -\sin(2\psi)]^T$, with $\psi$
the modulation angle of the $QWP$. After a few calculations, the dust backscattered light intensity $I_\lambda$ at wavelength $\lambda$ is
calculated as shown in Eq. 6:

$\quad I_\lambda(\psi) = I_{\lambda,0} \times [a_\lambda - b_\lambda \cos(4\psi)]$ (6)

where the intensity $I_{\lambda,0} = \eta_\lambda P_\lambda/(4d^2)$, while coefficients $a_\lambda$ and $b_\lambda$ are equal to $a_\lambda = F_{11,\lambda} + F_{22,\lambda}$ and $b_\lambda = 3F_{22,\lambda} - F_{11,\lambda}$.
Hence, $F_{22,\lambda}/F_{11,\lambda} = (1 + b_\lambda/a_\lambda)/(3 - b_\lambda/a_\lambda)$ so that the ratio $F_{22,\lambda}/F_{11,\lambda}$ at $\pi$−angle can be determined from the ratio
$b_\lambda/a_\lambda$. This ratio can be obtained from measurements of $I_\lambda(\psi)$, for different $\psi$-angles of the QWP, then adjusting these
variations with Eq. (6) to get accurate determinations of $I_{\lambda,0}a_\lambda$ and $I_{\lambda,0}b_\lambda$, then $b_\lambda/a_\lambda$. Evaluations of the dust lidar *PDR* are
then finally retrieved from Eq. (4):

$PDR_\lambda = (1 - b_\lambda/a_\lambda)/2$            (7)

Within our methodology, the dust lidar *PDR* is independent of $I_{\lambda,0}$. For that reason, in Section 4, the applied voltage to the UV
and VIS-photodetectors is adjusted to each dust $SD$ and mineralogy to gain in accuracy in the retrieved dust lidar *PDR* by
improving the signal-to-noise ratio on $I_\lambda$. For example, Fig. 3 provides simulations of $I_\lambda(\psi)/I_{\lambda,0}$ for the three following dust
lidar *PDR* case studies : 33 % dust lidar *PDR* (in full lines, i.e. $F_{22,\lambda}/F_{11,\lambda} = 0.5$), 25 % dust lidar *PDR* (in dashed-lines, i.e.
$F_{22,\lambda}/F_{11,\lambda} = 0.6$), 10 % dust lidar *PDR* (in dotted lines, i.e. $F_{22,\lambda}/F_{11,\lambda} = 0.82$). The curve minima, which are equal to
$I_{\lambda,m}/I_{\lambda,0} = a_\lambda - b_\lambda = F_{11,\lambda} - F_{22,\lambda}$, are shape-dependent : each curve hence exhibits non-vanishing minima since mineral dust
particles are nonspherical. Likewise, the curve maxima are equal to $I_{\lambda,M}/I_{\lambda,0} = a_\lambda + b_\lambda = 2F_{22,\lambda}$ and are size-dependent,
though it is also shape dependent. The dust lidar *PDR* is determined from $I_{\lambda,m}$ and $I_{\lambda,M}$ since, following Eq. (7),
$PDR_\lambda = I_{\lambda,m}/(I_{\lambda,m} + I_{\lambda,M})$, independently of $I_{\lambda,0}$.

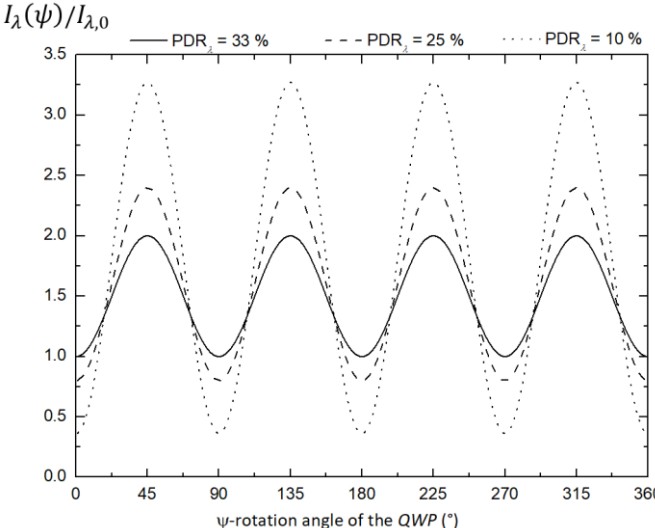


Figure 3: Numerical simulation of the dust backscattered light intensity $I_\lambda(\psi)/I_{\lambda,0}$ as a function of the orientation $\psi$ of the $QWP$ at a given
wavelength at the three following case studies: $PDR_\lambda = 33$ % (in full lines, corresponding to $F_{22,\lambda}/F_{11,\lambda} = 0.50$), : $PDR_\lambda = 25$ % (in
dashed-lines, $F_{22,\lambda}/F_{11,\lambda} = 0.60$), : $PDR_\lambda = 10$ % (in dotted lines, $F_{22,\lambda}/F_{11,\lambda} = 0.82$).

## 3.5 Accuracy on the retrieved laboratory mineral dust lidar PDR

Special care has been taken to quantify the uncertainties on the retrieved dust lidar $PDR$. The systematic errors in the $\pi$-polarimeter are that encountered in $2\lambda$-polarization lidar experiments, which we extensively studied (David et al., 2012) and can also be found in polarization lidar reference papers (Freudenthaler, 2016). To summarize, systematic errors arise from:

- *Imperfect definition of the polarization state of the incident radiation.* In the $\pi$–polarimeter, the polarization state of the electromagnetic radiation emitted from the laser is precisely set to $[1, 1, 0, 0]^T$ (i.e. with no remaining ellipticity) by using two successive $PBC$.

- *Polarization cross-talk between the emitter and the detector polarization axes.* Likewise, on the detector side, to account for the imperfections of the retro-reflecting $PBC$ ($R_s > 99.5\,\%, T_p > 90\,\%$), a secondary $PBC$ is inserted between the retro-reflecting $PBC$ and the light detector to ensure polarization cross-talk or undesired fraction $R_p T_s$ originating from the $p$-component of the backscattered radiation are fully negligible. Hence, the $\pi$–polarimeter is sensitive to the $s$-component of the backscattered radiation only. Also, the emitting $PBC$ being used as retro-reflecting PBC, any possible mismatch between the $s$-polarization axis of the emitted and detected backscattered radiations cannot occur.

- *Spectral cross-talk between the UV and the VIS-backscattered radiations.* Likewise, wavelength cross-talk is minimized by using selective interference filters exhibiting a higher than 5 optical density, at 355 nm wavelength in the VIS $\pi$–polarimeter and at 532 nm wavelength in the UV $\pi$–polarimeter.

- *Multiple scattering can induce further light depolarization.* However, the single-scattering approximation is ensured in our laboratory backscattering experiment (Mishchenko et al., 2007) where the particles are moving in a thin (2.5 mm) wide beam so that the volume element is optically thin in contrary to atmospheric chambers (1100 cm$^{-3}$ for the coarser SD).

Finally, to account for potential fluctuations in the dust particle number concentration that may cause variations in the dust backscattered light intensity $I_\lambda$, a normalization channel has been added to the $\pi$-polarimeter by including a polarization-insensitive light detector operating at scattering angle $\theta_0 = 165°$. The corresponding scattered light intensity $I_\lambda(\theta_0)$ is quantified similarly to Eq. 5 considering a scattering angle of $\theta_0 : I_\lambda(\theta_0) = [1, 0, 0, 0]^T[\mathbf{F}_\lambda(\theta_0)][\mathbf{QWP}(\psi)][\mathbf{PBC}][1, 1, 0, 0]^T$, where $[\mathbf{F}_\lambda(\theta_0)]$ is the scattering matrix at angle $\theta_0$. There, the $QWP$ and the $PBC$ only act on the detector side while $(St_i)$ equals $[1, 1, 0, 0]^T$. Hence, $I_\lambda(\theta_0) = I_{\lambda,0} \times [2F_{11,\lambda}(\theta_0) + F_{12,\lambda}(\theta_0) + F_{12,\lambda}(\theta_0)\cos(4\psi)]$. Once the variations of $I_\lambda(\theta_0)$ with $\psi$-angle are recorded, the $\cos(4\psi)$-dependency of $I_\lambda(\theta_0)$ can be removed by applying a numerical low-pass filter on $I_\lambda(\theta_0)$, to get a light intensity proportional to the dust particles number concentration. As a result, in the light backscattering curves presented in Section 4, the plotted quantity is the normalized backscattered light intensity $I_{\lambda,N} = I_\lambda(\pi)/I_\lambda(\theta_0)$, which is insensitive to potential fluctuations in the dust particles number concentration. The scattered light intensities $I_\lambda(\pi)$ and

$I_\lambda(\theta_0)$ being correlated, the standard deviation $\sigma_N$ on $I_{\lambda,N}$ was calculated by considering the covariance $\sigma_{I_\lambda I_\lambda(\theta_0)}$ of $I_\lambda$ and
$I_\lambda(\theta_0)$. This covariance contributes to the uncertainty on $I_{\lambda,N}$ at a rate $-2I_\lambda \sigma_{I_\lambda I_\lambda(\theta_0)}/I_\lambda^3(\theta_0)$. Moreover, to gain in accuracy in
the dust lidar $PDR$ retrievals, $I_{\lambda,N}$ was measured for a complete $\psi$-angle rotation, while averaging the acquired backscattered
light intensity over several thousand laser shots per $\psi$-angle, with resulting mean and standard deviations on $I_{\lambda,N}$ as plotted in
Fig. 4 and 5.
**4. Results and discussion**
In this section, using the methodology presented in Section 3, the lidar $PDR$ of Arizona dust, Asian dust, silica and hematite
is evaluated and discussed at 355 and 532 nm wavelength for the finer and the coarser $SD$.
**4.1 Laboratory evaluation of the lidar $PDR$ of Arizona and Asian dust**
Figure 4 displays the variations of $I_{\lambda,N}$ for Arizona (Fig. $4a$) and Asian dust (Fig. $4b$) as a function of the $\psi$-rotation angle
of the $QWP$ for the finer (left panels) and the coarser $SD$ (right panels) at 355 and 532 nm wavelength. The observed variations
are related to a determined size and shape distribution of the dust sample: indeed, as explained in Section 3.4, if the size (resp.
the shape) of the dust sample was varying during our acquisitions, the maxima (resp. the minima) of the curves would not
remain constant. As a result, the observed variations of $I_{\lambda,N}$ reveal the spectral and polarimetric backscattering characteristics
of each considered dust sample. Therefore, the experimental data points could be fitted with Eq. (6) to evaluate $F_{22,\lambda}/F_{11,\lambda}$
then the dust lidar $PDR$ by applying Eq. (7). Table 1 presents the retrieved values of $F_{22,\lambda}/F_{11,\lambda}$ and of dust lidar $PDR$. The
uncertainty on $F_{22,\lambda}/F_{11,\lambda}$ results from the measurement errors of the laboratory $\pi$-polarimeter and leads to accurate evaluations
of the dust lidar $PDR$. Within experimental error bars, the lidar $PDR$ of Arizona and Asian dust clearly differ, whatever the
chosen wavelength. The generally admitted value of around 33 % for the dust lidar $PDR$ (Tesche et al., 2009) is only obtained
for Arizona dust: Asian dust exhibits a lower $PDR$ in the range from 24 to 28 % depending on the considered $SD$ and
wavelength. This suggests that the dust lidar $PDR$ is primarily governed by the dust mineralogy and hence particles refractive
index. The sensitivity of the dust lidar $PDR$ with the considered $SD$ is indeed less pronounced: from the coarser to the finer
$SD$, a reduction in the dust lidar $PDR$ of below 5 % is observed at 532 nm wavelength. At 355 nm wavelength however, the
Arizona and Asian dust lidar $PDR$ seem practically insensitive to variations in the considered $SD$.

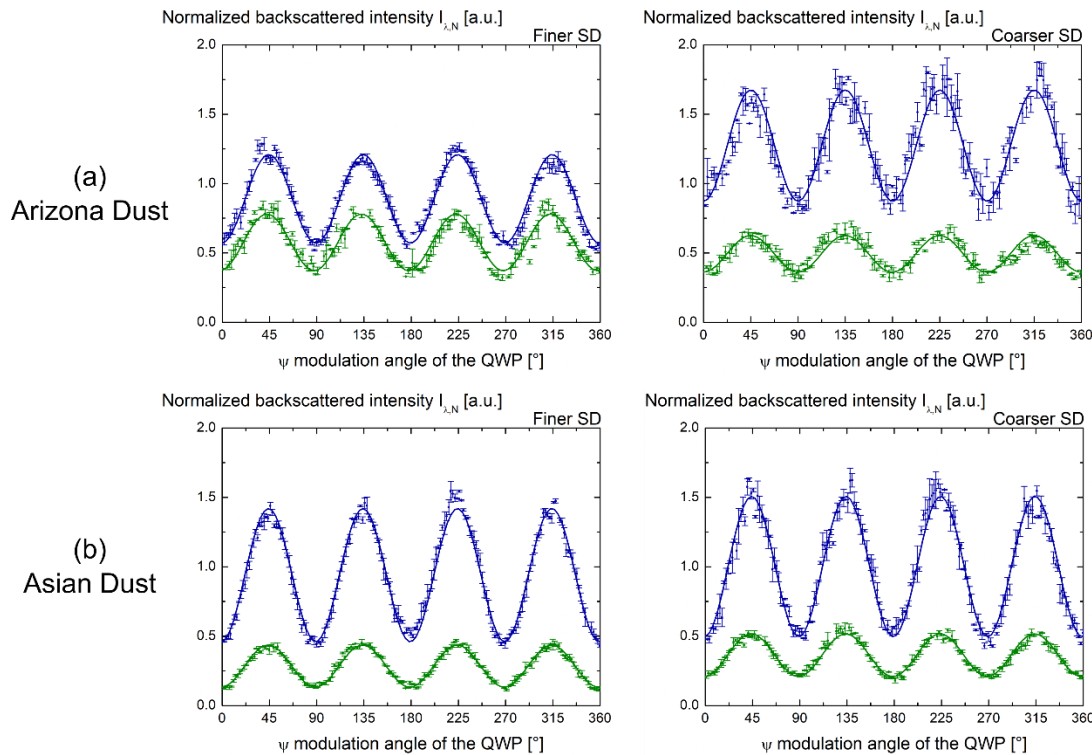


Figure 4: Normalized backscattered light intensity $I_{\lambda,N} = I_\lambda(\pi)/I_\lambda(\theta_0)$ of Arizona (a) and Asian dust (b) for finer $SD$ (left panels) and
coarser $SD$ (right panels), using the laboratory $\pi$-polarimeter at lidar exact backscattering angle ($\theta = \pi$) at 355 (blue) and 532 (green) nm.
The experimental data points are fitted with Eq. 6 to derive $F_{22,\lambda}/F_{11,\lambda}$ and then the dust lidar $PDR$ is derived using Eq. 7. Care should be
taken when comparing $I_{\lambda,N}$ for Arizona and Asian dust since the applied voltage to the UV and VIS-photodetectors was adjusted to increase
the signal-to-noise ratio, as explained in Section 3.4. The Arizona dust lidar $PDR$, retrieved from $I_{\lambda,m}/(I_{\lambda,m} + I_{\lambda,M})$, is higher than that of
Asian dust.
Tab. 1: Laboratory measurement of the $PDR$ of Arizona and Asian dust at 355 (blue) and 532 nm (green), for the finer and the coarser $SD$.
The $PDR$ is calculated with Eq. 7 after the derivation of $F_{22,\lambda}/F_{11,\lambda}$ using the laboratory $\pi$-polarimeter presented in Section 3.2. The
uncertainty on $F_{22,\lambda}/F_{11,\lambda}$ is deduced from the evaluation of $b_\lambda/a_\lambda$, itself deduced from the least-square fit adjustment of $I_\lambda$. The uncertainty
on $F_{22,\lambda}/F_{11,\lambda}$ is mostly dominated by statistical uncertainties since our biases are minimized, as explained in Section 3.5.

| Mineralogy | $\lambda$ | Finer $SD$ | | Coarser $SD$ | |
|---|---|---|---|---|---|
| | $(nm)$ | $F_{22,\lambda}/F_{11,\lambda}$ | $PDR_\lambda$ (%) | $F_{22,\lambda}/F_{11,\lambda}$ | $PDR_\lambda$ (%) |
| Arizona dust | 355 | $0.514 \pm 0.007$ | $\mathbf{32.1 \pm 0.6}$ | $0.489 \pm 0.012$ | $\mathbf{34.3 \pm 1.0}$ |
| | 532 | $0.512 \pm 0.012$ | $\mathbf{32.3 \pm 1.0}$ | $0.464 \pm 0.012$ | $\mathbf{36.6 \pm 1.1}$ |
| Asian dust | 355 | $0.603 \pm 0.009$ | $\mathbf{24.7 \pm 0.6}$ | $0.603 \pm 0.011$ | $\mathbf{24.8 \pm 0.8}$ |
| | 532 | $0.622 \pm 0.009$ | $\mathbf{23.3 \pm 0.7}$ | $0.558 \pm 0.011$ | $\mathbf{28.4 \pm 0.8}$ |


**4.2 Laboratory evaluation of the lidar *PDR* of silica and hematite**
By applying the same methodology, we obtain the *PDR* of silica and hematite, as presented in Fig. 5 and Table 2.
Accordingly, Fig. 5 is the analog of Fig. 4 for silica (Fig. 5a) and hematite (Fig. 5b). As for Arizona and Asian dust samples,
the lidar *PDR* of silica and hematite primarily depends on the particles *CRI*, at least at 355 nm wavelength where the silica
lidar *PDR* ranges from 23 to 33 % depending on the considered *SD* while the hematite lidar *PDR* reaches 10 % only. The silica
lidar *PDR* also strongly depends on the particles diameter: from the coarser to the finer SD, the silica dust lidar PDR reduces
by 10 % at both wavelengths. The dependence of the hematite dust lidar *PDR* with the *SD* is less pronounced, especially at
355 nm wavelength. The silica and hematite lidar *PDR* also strongly depend on the chosen lidar wavelength, with higher
depolarization observed at 355 nm wavelength for silica and at 532 nm wavelength for hematite.

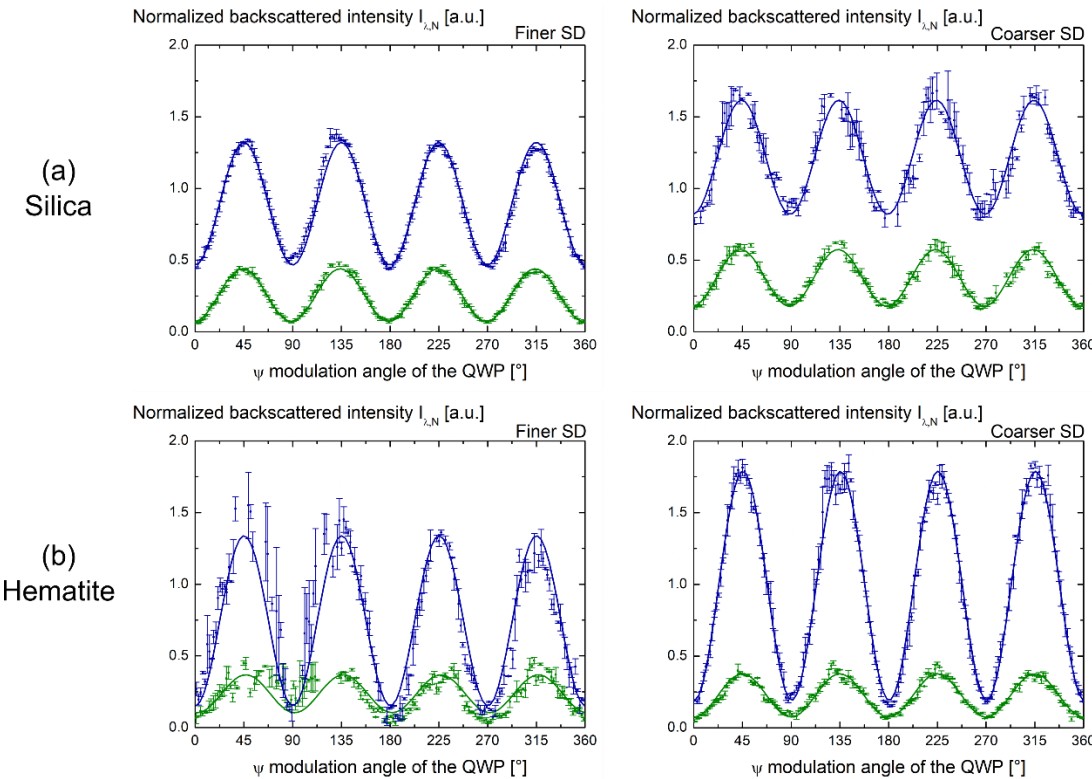


Figure 5: Same as Fig. 4 for silica (a-plots) and for hematite samples (b-plots).


Tab. 2: Same as Table 1 for silica and hematite.

| Mineralogy | $\lambda$ | Finer $SD$ | | Coarser $SD$ | |
|---|---|---|---|---|---|
| | $(nm)$ | $F_{22,\lambda}/F_{11,\lambda}$ | $PDR_\lambda$ (%) | $F_{22,\lambda}/F_{11,\lambda}$ | $PDR_\lambda$ (%) |
| Silica | 355 | $0.622 \pm 0.014$ | $\mathbf{23.3 \pm 0.9}$ | $0.506 \pm 0.011$ | $\mathbf{32.8 \pm 1.0}$ |
| | 532 | $0.751 \pm 0.016$ | $\mathbf{14.2 \pm 0.9}$ | $0.618 \pm 0.016$ | $\mathbf{23.6 \pm 1.1}$ |
| Hematite | 355 | $0.805 \pm 0.050$ | $\mathbf{10.8 \pm 2.5}$ | $0.823 \pm 0.015$ | $\mathbf{9.7 \pm 0.7}$ |
| | 532 | $0.652 \pm 0.055$ | $\mathbf{21.1 \pm 3.5}$ | $0.715 \pm 0.019$ | $\mathbf{16.6 \pm 1.1}$ |


## 4.3 Discussion

Comparing our laboratory findings with other laboratory experiments is not straightforward, since as explained in the
introduction, none operates at 180.0° lidar exact backscattering angle, while the dust lidar $PDR$ differs from near to exact
backscattering angles, especially when light absorbents are present (Cholleton et al., 2022). Moreover, the $PDR$ is wavelength-
dependent and the size distributions ($SD$) used are different from other studies. Lidar field experiments provide accurate values
of the lidar $PDR$ after accurate calibration procedure based on the scattering matrix (Freudenthaler, 2016; Belegante et al.,
2018; Miffre et al., 2019). Although in such lidar field experiments, the measured $PDR$ is usually that of dust mixtures (Miffre
et al., 2011), the comparison with our laboratory findings remains interesting. In lidar retrievals (see for example (Tesche et
al., 2009)), a dust lidar $PDR$ of 30 % is often used. The laboratory $\pi$-polarimeter verifies this statement by providing the silica
$PDR$, which is the main oxide present in mineral dust, equal to $(33 \pm 1)$ % for the coarser $SD$ at 355 nm. In comparison,
within our experimental error bars, the hematite lidar $PDR$, equal to $(10 \pm 1)$ %, is clearly lower. The real part $n$ and the
imaginary part $\kappa$ of the hematite $CRI$, which are large compared with that of other chemical oxides present in mineral dust (see
Section 2.1), can be responsible for the observed difference in the silica and hematite lidar $PDR$. Indeed, $n$ and part $\kappa$ modify
the backscattering matrix elements, so does the corresponding dust lidar $PDR$. To highlight the role of $\kappa$ on the hematite lidar
$PDR$, the lidar $PDR$ of rutile was measured with our $\pi$-polarimeter. Indeed, the real part of the rutile $CRI$ is as large as that of
hematite but its imaginary part is negligible compared with that of hematite. As a result, the rutile lidar $PDR$ substantially
differed from that of hematite, showing the key role played by light absorption in the measured hematite lidar $PDR$. In turn,
Arizona dust exhibits a higher $PDR$ than Asian dust, due to the higher proportion in hematite in the latter. Hence and as a
conclusion, our laboratory findings show that, when the light absorbent hematite is present, it mainly governs the dust lidar
$PDR$, which hence primarily depends on the particles mineralogy, with less pronounced variations with the particles size and
wavelength. This finding is in line with (Kahnert, 2015; Kahnert et al., 2020) numerical findings, who highlighted that the dust
$PDR$ is strongly modulated by the particles inhomogeneity, especially in the lidar backward scattering direction and in the
presence of hematite. We here quantify this effect with a laboratory experiment that accounts for the real shape of mineral
dust. The shape dependence of the hematite PDR is weak due to its large imaginary part of complex refractive index: following
Wiscombe and Mugnai (1986) or Mishchenko et al. (1997), the effect of particle shape becomes weaker with increasing
imaginary part of the refractive index, a conclusion also drawn by Meland et al. (2011). In contrast, when the proportion of
hematite becomes negligible, as is the case for silica and Arizona dust, our laboratory findings show that the dust lidar $PDR$
then increases with increasing the particles size, though the shape dependence may then also play a role. Also, it would be
interesting to investigate giant dust particles (Ryder et al., 2019). Likewise, in the literature (Sakai et al., 2010; Hofer et al.,
2020; Järvinen et al., 2016; Mamouri and Ansmann, 2017), the dust lidar $PDR$ is usually found to increase with the particles
size from the fine to the coarse mode of the $SD$. The (355, 532) nm wavelength dependence of the dust lidar $PDR$ then becomes
key for discussing on the involved particle sizes, thus underlying the importance of dual-wavelength (or more) polarization
lidar instruments. We here establish this result in laboratory at 180.0° and (355, 532) nm wavelength, and moreover, show that
this consideration holds only when hematite, which is a strong light absorbent, is not involved : the hematite lidar $PDR$ is
indeed higher in the finer mode of the $SD$.

To go further and discuss on the role of light absorption in the retrieved dust lidar $PDR$, we here propose a basic partitioning
model in which the dust particles mixture $(d) = \{Abs, \overline{Abs}\}$ is comprised of two components: an absorbing component $(Abs)$,
mainly corresponding to hematite particles, and a non-absorbing component $(\overline{Abs})$, mainly corresponding to silica-particles.
For simplicity, we here resume the absorbing (resp. non-absorbing) component to hematite (resp. silica)-particles with
respective abbreviations $(Hmt)$ and $(Sil)$. We focus on the 355 nm wavelength at which hematite is an efficient light absorber
and on the coarser $SD$ as the dependence of the dust lidar $PDR$ with size is less pronounced than with the particles mineralogy.
In Appendix A is detailed the derivation of the lidar $PDR$ of such a dust-particles mixture $(d) = \{Hmt, Sil\}$ (hereafter noted
$\delta_d$, as in lidar applications). This Appendix is an extension of our previous works (Miffre et al., 2011; David et al., 2013, 2014;
Mehri et al., 2018) for the case study where both components $\{Hmt, Sil\}$ are nonspherical. The lidar $PDR$ of such a dust-
particles mixture relates to that of its pure components (hereafter noted $\delta_{Sil}$ and $\delta_{Hmt}$) as follows:

$$\delta_d = \frac{-e + (c + e)X_{\text{Hmt}}}{f - (d + f)X_{\text{Hmt}}} \qquad\qquad (8)$$

where the expressions of the $c, d, e$ and $f$-coefficients are provided in Appendix A and independently on the depolarization
ratios $\delta_{Sil}$ and $\delta_{Hmt}$ of silica and iron oxides. $X_{Hmt}$ is the fraction of $Hmt$ to dust particles backscattering. Following Eq. (8)
and Appendix A, Fig. 6 displays the variation of $\delta_d$ as a function of $X_{Hmt}$ when considering $\delta_{Sil}$ = 33 % and $\delta_{Hmt}$ = 10 %, as
obtained in our laboratory findings at 355 nm wavelength with the coarser $SD$. As shown in Fig. 6, the dust lidar $PDR$ lies in
between $\delta_{Sil}$ and $\delta_{Hmt}$ and equals $\delta_{Sil}$ (resp. $\delta_{Hmt}$) only when $X_{Hmt}$ = 0 (resp. 1), depending on the fraction $X_{Hmt}$ of light
corresponding to the absorbent of the dust particle mixture. Hence, Arizona dust, which contains a lower fraction of hematite,
exhibits a higher lidar $PDR$ compared with Asian dust, at least at 355 nm wavelength where hematite is strongly absorbing.
Though rather simple, our model interestingly highlights the key role played by light absorption in the retrieved Asian dust
lidar $PDR$. To go further and provide a quantitative analysis, this simple model should be refined, by considering also the other
chemical oxides present in mineral dust, other lidar wavelengths, as well as other SD and the effect of shape. To handle such
a complex issue, more laboratory experiments are required on other chemical oxides, ideally also at 1064 nm wavelength. This
work is however beyond the scope of this paper. Still as is, our model provides an interpretation of the laboratory-observed
differences in the dust lidar $PDR$ when the light absorbent hematite is involved. In the most general case, the dust lidar $PDR$
hence appears as a complex function of the particles mineralogy, $SD$, wavelength and shape. Comparison with lidar field
experiments, involving particle mixtures, with a more complex distribution of sizes and refractive indices, is then not
straightforward, as underscored by comparison with Hu et al. (2020) who reported 0.28 - 0.32 ± 0.07 at 355 nm wavelength.
Though this complex dependence is difficult to disentangle, our laboratory findings show that the dust lidar $PDR$ is primarily
affected by the particles mineralogy, at least when hematite is involved.

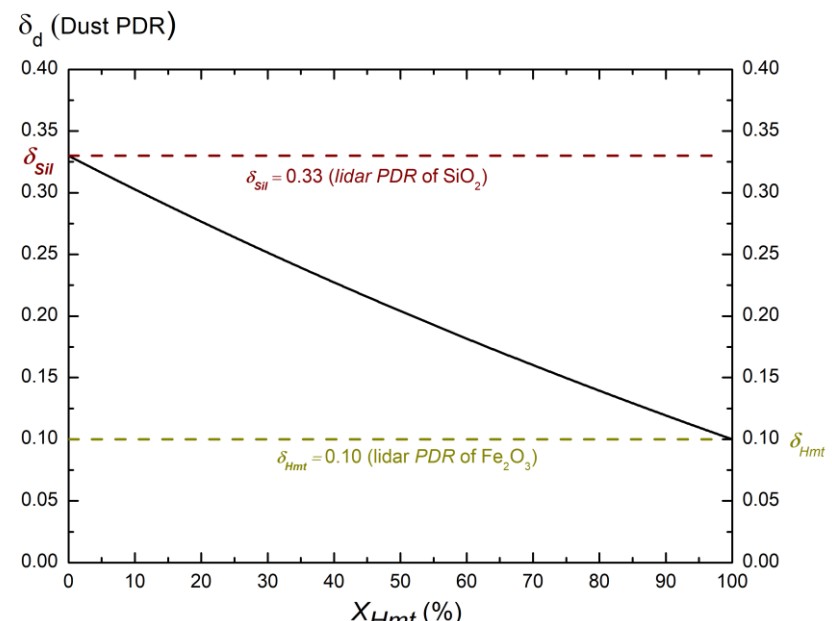


Figure 6: Numerical simulation of the 355 nm Lidar $PDR$ of a two-component particles mixture $(d) = \{Hmt, Sil\}$, composed of hematite
($Hmt$) and silica ($Sil$) oxides as a function of the $X_{Hmt} = \beta_{Hmt}/\beta_d$ fraction of $Hmt$ to $d$-particles backscattering, following Eq. (8) and
Appendix A, by accounting for our laboratory experimental findings for $\delta_{Sil} = 33$ % and $\delta_{Hmt} = 10$ % (see Table 2 at 355 nm wavelength
with coarser $SD$).

## 5 Summary and conclusion

In this paper, the dependence of the lidar particles depolarization ratio ($PDR$) of pure mineral dust with complex refractive index ($CRI$) and size is for the first time investigated through a laboratory $\pi$-polarimeter operating at 180.0° lidar backscattering angle and (355, 532) nm wavelengths for lidar purposes. The goal of this work is to improve the knowledge on the dust lidar $PDR$, which is an important input parameter involved in lidar partitioning algorithms, which are widely applied to reveal the contribution of mineral dust in particles external mixtures (Tesche et al., 2009; Mehri et al., 2018). While mineral dust exhibits a complex and highly irregular shape, which is difficult to model mathematically and numerically, our laboratory approach allows accounting for the real shape of mineral dust. Our laboratory $\pi$-polarimeter is likewise a good complement to lidar field experiments, which provide accurate retrievals of the lidar $PDR$ of particles mixtures involving mineral dust. Another advantage of our laboratory $\pi$-polarimeter lies in its ability to provide accurate retrievals of the lidar $PDR$ of pure mineral dust samples, differing in $CRI$ and size. The $\pi$-polarimeter indeed operates at 180.0° lidar backscattering angle and at (355, 532) nm lidar wavelengths: no assumption is made to retrieve the dust lidar $PDR$. This is a key novelty of our study. Indeed, the variation of the dust lidar $PDR$ with scattering angle and wavelength cannot be analytically calculated (Bohren and Huffman, 1983; Mishchenko et al., 2002) for complex-shaped particles such as mineral dust. Hence, our $\pi$-polarimeter improves the knowledge on the dust $PDR$, provided in the literature at non 180.0° backscattering angle and / or at wavelengths differing from (355, 532 nm). Our work provides sixteen accurate dust lidar $PDR$-values, corresponding to four different complex refractive indices, studied at two size distributions (fine, coarse) and at (355, 532) nm wavelengths (see Section 4). The precision on the retrieved dust $PDR$ from the laboratory $\pi$-polarimeter is detailed in Section 3. To investigate the dependence of the dust lidar $PDR$ with $CRI$, hematite, the main light absorbent present in mineral dust, was considered in addition to silica oxide, the main chemical oxide present in mineral dust, which is practically nonabsorbent. At 355 nm, our laboratory $\pi$-polarimeter provides values of the $PDR$ of coarser silica of $(33 \pm 1)$ % while that of coarser hematite is only $(10 \pm 1)$ %. In Section 4, this large difference is explained by accounting for the high imaginary part of the hematite $CRI$. In turn, Arizona dust exhibits a higher depolarization ratio than Asian dust, due to the higher proportion in hematite in the latter. As a result, when the strong light absorbent hematite is involved, the dust lidar $PDR$ is primarily governed by the particles mineralogy and the variations of the dust lidar $PDR$ with size are less pronounced. The dependence of the PLDR on the particles shape is not pronounced in our experiment where hematite, which exhibits a large imaginary part of complex refractive index, plays a key role (Wiscombe and Mugnai, 1986, Mishchenko et al., 1997, Meland et al., 2011). When hematite is less or not involved, the dust lidar $PDR$ increases with increasing sizes and the (355, 532) nm wavelength dependence of the dust lidar $PDR$ then becomes key for discussing on the involved particle sizes, thus underscoring the importance of dual wavelengths (or more) polarization lidar instruments. To further disentangle the complex dependence of the dust lidar $PDR$ with complex refractive index and size, our methodology should be extended to other chemical oxides, other natural mineral dust samples, other $SD$ and other wavelengths, as well as other shape distributions. Giant dust particles, whose importance has been highlighted by Ryder et al. (2019), would likewise be interesting to study specifically. This is however far beyond the

scope of this paper : we here focused on (355, 532) nm wavelengths, since mineral dust slightly absorb light in the near infra-
red (Di Biagio et al., 2019). Still, the above laboratory findings underscore the importance of accounting for the wavelength
dependence of the dust lidar $PDR$, whatever the hematite proportion. The spectral dependence of the dust lidar $PDR$ is indeed
instructive (Burton et al., 2016; Haarig et al., 2022; Miffre et al., 2020). Outlooks of this work are obviously also interesting,
as underscored by recent papers (Kahnert et al., 2020; Luo et al., 2022), discussing on the ability of the spheroidal model to
mimic light scattering by complex-shaped mineral dust.
**Appendix A**
The goal of this Appendix is to establish the expression of the lidar $PDR$ of a two-component particle mixture $(p) =$
$\{ns_1, ns_2\}$ composed of two non-spherical components $ns_1$ and $ns_2$. As in lidar applications, the lidar $PDR$ of $p, ns_1$ and $ns_2$-
particles are respectively noted $\delta_p, \delta_{ns_1}$ and $\delta_{ns_2}$. The starting point is given by the set of four equations:

$\beta_{p,//} = \beta_{ns_1,//} + \beta_{ns_2,//}$ (A-1-a)
$\beta_{p,\perp} = \beta_{ns_1,\perp} + \beta_{ns_2,\perp}$ (A-1-b)
$\delta_{ns_1} = \beta_{ns_1,\perp}/\beta_{ns_1,//}$ (A-1-c)
$\delta_{ns_2} = \beta_{ns_2,\perp}/\beta_{ns_2,//}$ (A-1-d)

where $\beta_{p,//}$ and $\beta_{p,\perp}$ are the lidar particles backscattering coefficients, evaluated from a polarization lidar experiment carried
out at wavelength $\lambda$ (here omitted to ease the reading). The backscattering coefficient $\beta_{ns_1}$ of $ns_1$-particles is then retrieved by
noting that $\beta_{ns_1} = \beta_{ns_1,//} + \beta_{ns_1,\perp} = \beta_{ns1,\perp}(1 + 1/\delta_{ns_1})$ (Miffre et al., 2011; David et al., 2013). Moreover, $\beta_{ns_1,\perp}$ can be
expressed as a fonction of $\beta_{p,//}$ and $\beta_{p,\perp}$ since $\beta_{ns_1,\perp} = \beta_{p,\perp} - \beta_{ns_2,\perp} = \beta_{p,\perp} - \delta_{ns_2}\beta_{ns_2,//} = \beta_{p,\perp} - \delta_{ns_2}(\beta_{p,//} - \beta_{ns_1,\perp}/\delta_{ns_1})$
using Eqs. (A-1). Hence, $\beta_{ns_1,\perp} = (\beta_{p,\perp} - \delta_{ns_2}\beta_{p,//})/(1 - \delta_{ns_2}/\delta_{ns_1})$. By applying the same methodology to $ns_2$- particles,
we finally get:

$\begin{pmatrix} \beta_{ns_1} \\ \beta_{ns_2} \end{pmatrix} = \begin{bmatrix} c & d \\ e & f \end{bmatrix} \begin{pmatrix} \beta_{p,//} \\ \beta_{p,\perp} \end{pmatrix}$ (A-2)

where the $c, d, e$ and $f$-coefficients only depend on the depolarization ratios $\delta_{ns_1}$ and $\delta_{ns_2}$ :

$c = -\delta_{ns_2}(1 + 1/\delta_{ns_1})/(1 - \delta_{ns_2}/\delta_{ns_1})$ (A-3-a)
$d = (1 + 1/\delta_{ns_1})/(1 - \delta_{ns_2}/\delta_{ns_1})$ (A-3-b)
$e = -\delta_{ns_1}(1 + 1/\delta_{ns_2})/(1 - \delta_{ns_1}/\delta_{ns_2})$ (A-3-c)
$f = (1 + 1/\delta_{ns_2})/(1 - \delta_{ns_1}/\delta_{ns_2})$                                                       (A-3-d)

The 2 x 2 matrix introduced in Eq. (A-2) can be inverted to get the expression of $\beta_{p,//}$ and $\beta_{p,\perp}$ and hence $\delta_p = \beta_{p,\perp}/\beta_{p,//}$. By
introducing $X_{ns2} = \beta_{ns2}/(\beta_{ns1} + \beta_{ns2})$ the fraction of $ns_2$ to $p$-particles backscattering, we finally get the relationship
between $\delta_p$ and $\delta_{ns1}$ and $\delta_{ns2}$ :

$\delta_p = \dfrac{-e + (c + e)X_{ns2}}{f - (d + f)X_{ns2}}$                                                     (A − 4)

In the specific case where $ns_2$-particles are spherical (i.e. $\delta_{ns_2} = 0$), the expressions of the $c, d, e$ and $f$-coefficients simplify
and the relationship between $\delta_p$ and $X_{ns2} = X_{ns}$ becomes identical to that we already published in (Miffre et al., 2011; David
et al., 2013). This new material is hence as an extension of our previous works (Miffre et al., 2011; David et al., 2013, 2014;
Mehri et al., 2018) to the case study where both components of the particles mixture $(p) = \{ns_1, ns_2\}$ are nonspherical.

**Author contribution**

**Alain Miffre**: Conceptualization, Formal analysis, Investigation, Methodology, Supervision, Writing - original draft, Writing
- review & editing **Danaël Cholleton**: Formal analysis, Investigation, Software, Visualization, Writing - review & editing.
**Clément Noël**: Software, Writing - review & editing **Patrick Rairoux**: Project administration, Supervision, Writing - review
& editing.

**Competing interests**

The authors declare that they have no conflict of interest.

**Acknowledgements**

The French National Center for Space Studies (CNES) is acknowledged for financial support.

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
