# Peer review of "Investigating the dependence of mineral dust depolarization on"

_Atmospheric Measurement Techniques, 2022_

## Author Comment (AC1)

Investigating the dependence of mineral dust depolarization on complex refractive index and size with a laboratory polarimeter at 180.0° lidar backscattering angle
Manuscript by A. Miffre, D. Cholleton, C. Noël and P. Rairoux
Submitted to Atmospheric Measurement Techniques (AMT).

We thank both reviewers for the time they spent to review our manuscript. The provided comments add value to our manuscript and may help future readers and we are grateful for that. For the sake of clarity, each comment is first recalled in italics, then our answer is given, together with a list of changes made to the manuscript, which also appear in red-lined version in the revised manuscript.

**Author's response to Reviewer #2 's comments**

*Reviewer #2:* *"This paper investigates the relationship between the particle depolarization ratio of mineral dust and the particles' complex refractive index and size, using a Pi-polarimeter that operated at a 180-degree backscattering angle, at two typical lidar wavelengths, 355 and 532 nm. Through laboratory experiments, the authors derive 16 dust-related particle depolarization ratio values that correspond to four different refractive indices (mineral dust samples with different mineralogy), for two size distributions (fine, coarse) and at two wavelengths (355, 532 nm).*

*The work falls well within the scope of AMT. Overall, the methodology is well explained and the results are clearly presented. However, the manuscript could be improved prior to publication, by addressing the comments provided below."*

We thank Reviewer #2 for the time she / he spent to review our manuscript. We are pleased to see that our work provides clearly presented results, in a well-explained methodology. For clarity, we answer point by point below to each comment. We thank Reviewer #2 for these improvements, which may help future readers. All the corrections made appear in red-lined version in the revised manuscript.

**Major comments:**

**Reviewer #2, comment 1:** *"The description of the dust samples (Sect. 2) should be more detailed. What exactly is Asian dust? Where do you get it from? Does it originate from a specific dessert? As I understand it, you use commercially available dust samples and silica and hematite as well. How are your four samples treated and prepared by the manufacturer? Furthermore, the paper would become richer, if the discussion about the mineralogical composition of desert dust samples could be added. There are various studies investigating the mineralogical composition including silica and hematite contributions. Apart from the finer/coarser SD differentiation (L144-L147), to which extent are the chosen dust samples representative/characteristic of what is being observed in the atmosphere?"*

> **Answer to Reviewer's 2 comment 1:** The Asian dust sample we use is provided by Powder Technology (commercial name: Kanto Loam). It was chosen for its large percentage in hematite (17-23 %), to investigate the role of the imaginary part of the refractive index on the PDR. It is commonly used as a test dust interferon in pollen light scattering measurements in Japan (Iwai et al., 2013), hence representative of observed atmospheric Asian dust. In this way, we symmetrized our approach by dealing with both Arizona Test Dust and Asian Test Dust. The information on the preparation of the samples is unfortunately not provided by the manufacturer. Nevertheless, the measured SD are representative of what is observed in atmosphere, with a low number concentration of more than 10 µm particles, as observed by Weizierl et al. (2017), and as detailed in our answer to Reviewer #1. Also, our SD correspond to that currently observed in other laboratory experiments (see Järvinen et al., 2016 for example).

The mineralogical composition is only given for hematite and silica, which are the main concern of our study. Other chemical oxides are also present in our dust samples in various percentages but their mineralogical composition is not detailed since they exhibit negligible imaginary parts of CRI compared with that of hematite. Also, as explained at lines 398-400 and 439 of our original manuscript, studying their PDR is an outlook of this work. To prepare this future work, we can of course add this information in % of Arizona Test Dust, then Asian Dust: $Al_2O_3$ (11 %, 29 %), CaO (4 %, 1.5 %), $K_2O$ (3.5 %, 0 %), $Na_2O$ (2 %, 0 %), MgO (1.5 %, 5 %), $TiO_2$ (0.5 %, 2 %). Thank you for your comment.

➢ **Changes made to the manuscript:** To account for Reviewer #2's comment, we added the following sentences to our revised manuscript:
- In Section 2.1, dedicated to the presentation of our dust samples: "For Asian dust, we use a commercial sample provided by Powder Technology (commercial name: Kanto Loam), commonly used as a dust interferon in pollen light scattering measurements in Japan (Iwai, 2013), hence representative of observed atmospheric Asian dust. In this way, we symmetrized our approach by dealing with both Arizona Test Dust and Asian Test Dust."
- The information on the mineralogical composition of our test dust samples was also added: "Other chemical oxides are also present in our dust samples in various percentages, but with negligible imaginary parts of CRI compared with that of hematite. Investigating the PDR of these oxides is then beyond the scope of this paper. Their percentage in (Arizona Test Dust, Asian Dust) is given for clarity: $Al_2O_3$ (11 %, 29 %), CaO (4 %, 1.5 %), $K_2O$ (3.5 %, 0 %), $Na_2O$ (2 %, 0 %), MgO (1.5 %, 5 %), $TiO_2$ (0.5 %, 2 %)."
- In Section 2.2, dedicated to the size of our dust samples: "The measured SD are representative of what is observed in atmosphere, with a low number concentration of more than 10 µm particles, as observed by Weizierl et al. (2017)".

**Reviewer #2, comment 2:** "*The size distributions (Fig. 1) are a finer and a coarser one as you often state, but it is not a fine mode and a coarse mode (as sometimes ambiguously stated, e.g., L12, L426). It is a fine mode size distribution and a fine + coarse mode size distribution or in other words a size distribution with and one without coarse mode. Please clearly make this statement in Sect. 2.2. »*

➢ **Answer to Reviewer's 2 comment 2:** Thank you for your comment, which adds precision to our manuscript. We agree and modified our manuscript accordingly.
➢ **Changes made to the manuscript:** We added this sentence to Section 2.2: "More precisely, the two considered SD correspond to a size distribution with and without coarse mode".

**Reviewer #2, comment 3:** "*How do you estimate the uncertainty of your results (Tab. 1 + 2)? Is it the uncertainty of the fit? To which amount does the systematic error is considered?*"

➢ **Answer to Reviewer's 2 comment 3:** Thank you for your comment. We indeed precisely evaluate the uncertainties in Tables 1 and 2 by applying the least-square fit method to the measured light backscattering light intensity. The systematic errors are minimized as explained in the beginning of Section 3.5, so that corresponding uncertainties are mainly due to statistical errors.
➢ **Changes made to the manuscript:** To account for Reviewer #2's comment, we added the two following sentences to the caption of Table 1: "The uncertainty on $F_{22,\lambda}/F_{11,\lambda}$ is deduced from the evaluation of $b_\lambda/a_\lambda$, itself deduced from the least-square fit adjustment of $I_\lambda$. The uncertainty on $F_{22,\lambda}/F_{11,\lambda}$ is mostly dominated by statistical uncertainties since our biases are minimized, as explained in Section 3.5."

**Reviewer #2, comment 4:** "*The discussion and comparison to previous literature is rather short and should be extended before publication. Even if previous laboratory setups did not operate at exactly 180°*

*backscatter, the results should be discussed. Especially, I am missing a reference and discussion to the work by Sakai et al., 2010, who investigated fine and coarse mode dust from Asia and the Sahara at 532 nm. How do their results compare to your new findings? The comparison to lidar field experiments is rather short as well. It is hard to compare for Arizona Test Dust, but for Asian dust, there are plenty of field experiments reporting PDR at 355 and/or 532 nm, e.g., Sugimoto & Lee, 2006; Hofer et al., 2020 or Hu et al., 2020."*

> **Answer to Reviewer's 2 comment 4:**
> Thank you for your comment. As you noted, comparison with other laboratory work at strict backscattering angle of 180.0° is not feasible since the scattering matrix elements may strongly vary over a small scattering angle range, especially when light absorbents are present (Cholleton et al., 2022). Otherwise, according to Eq. (3), a small variation in $F_{11,\lambda}$ and in $F_{22,\lambda}$ can result in a large variation of the lidar PDR. Though our samples are different, Sakai et al., (2010) retrieved increasing lidar PDR with size at 532 nm wavelength, what we also observe. We already compared our results with Hofer et al. (2020) at line 370 of our original manuscript and found an increase in the dust lidar PDR with particles size. We may precise the values: Hofer et al. (2020) reported lidar PDR ranging from 0.23 to 0.29 at 355 nm wavelength, in line of our laboratory results. The differences are more pronounced at 532 nm wavelength. Comparison with lidar field experiments, involving particle mixtures, with a more complex distribution of sizes and refractive indices, is however not straightforward, as underscored by comparison with Hu et al. (2020) who reported 0.28-0.32 ± 0.07) at 355 nm wavelength.
>
> **Changes made to the manuscript:** To account for Reviewer #2's comment, we added the above discussion to our revised manuscript and especially reference to Sakaï et al. (2010) at line 370 of the original manuscript where we emphasized that the PDR was increasing with particles sizes. We also added that "Comparison with lidar field experiments, involving particle mixtures, with a more complex distribution of sizes and refractive indices, is however not straightforward, as underscored by comparison with Hu et al. (2020) who reported 0.28-0.32 ± 0.07 at 355 nm wavelength."

**Minor comments:**

*"Please always state Arizona Test Dust and not just Arizona dust. Arizona Test Dust is a well-known term in the community."*
Thank you for your comment, we modified the manuscript accordingly.

*"L43-47: need rephrasing. Also, the literature selected is rather limited, important studies are missing."*
We will add reference to Hofer et al. (2020) and Hu et al. (2020) there in the revised manuscript.

*"L49: The particle linear depolarization ratio's importance for aerosol typing has been demonstrated in numerous studies (e.g., Burton et al., 2012). The authors should extend the literature provided here accordingly."*
We already quoted the paper by Burton et al. (2016), but we can of course also add that from 2012.

*"L70- 82: It would have been better if the authors merged the list with the main body text."*
We do not want to merge the list with the main body text as it is important to emphasize the novelty of our work compared with that of other existing laboratory experimental set-ups.

*"L102: new paragraph "The paper is structured…"*
We agree and did the correction.

*"L122-124: The imaginary part of the CRI varies by a factor of 10 between the literature values: 0.0925 (Longtin et al., 1988) and 0.9 or 0.6 (Go et al., 2022). Is there a reason for the difference?"*

Thank you for your comment. Triaud et al. (2005) is in agreement with Go et al. (2022). We will keep Go et al. (2022).

"*Just out of curiosity: Why do your size distributions (Fig. 1) all show a peak at 1 μm?*"
This is a good question: as we did not yet explore all size distributions, it is difficult to explain. It however remains representative of observed atmospheric mineral dust size distributions.

"*In line 267 you're referring to the polarization lidar reference paper of Freudenthaler et al., 2009. There is an even more complete assessment of the polarization lidar calibration given by the same author (Freudenthaler, 2016). There, additional sources of uncertainties are discussed. In your case, the rotational misalignment around the optical axis might be worth discussing (even if it is probably very small).*"
We will add reference to Freudenthaler et al. (2016) at line 267 of the revised manuscript. We also checked that the rotational misalignment around the optical axis is very small.

"*L282-284: Please provide an approximate particle concentration.*"
The concentration can be found be integration over the Figure 1 size distributions. We found $1.11 \times 10^3$ $cm^{-3}$ for the coarser SD and added the value to the manuscript.

"*Lidar particles depolarization ratio – lidar PDR: Does the term "lidar PDR" refers to the 180° backscatter direction? Or what is the difference to PDR?*"
Thank you for your comment. In Equations (3) and (4), we precisely emphasized the distinction between PDR and the lidar PDR, the latter being specifically dedicated to the 180.0° lidar backscattering angle.

"*At one instance, you should mention that you are measuring the linear depolarization ratio.*"
We agree and added the word linear at line 186 and the following sentence to Section 3.1: "The PDR stated in Equation (3) is the linear PDR, which can be related to the circular PDR if need be (Mishchenko et al., 2002)."

"*L351-353: Please rephrase. In field experiments, we do observe pure aerosol conditions with lidars- not only aerosol mixtures.*"
We agree and modified the sentence as follows : "Although in such lidar field experiments, the measured PDR is usually that of dust mixtures (Miffre et al., 2011), the comparison with our laboratory findings remains interesting."

"*Fig. 1: Larger fonts (for labels, markers, axis) are needed. Consider changing the grey colour, it is very hard to read.*"
Thank you for your comment, we did the proposed correction.

[Figure]

"*Fig. 4: Larger fonts are needed. Very hard to read. There is enough space in the plot to include the names of the dust samples (Arizona Test, Asian). The same holds for Fig. 5.*"

Thank you for your comment, we did the proposed correction.

[Figure]

(a)
Arizona Dust

(b)
Asian Dust

(a)
Silica

(b)
Hematite

"*Fig. 6: It would be recommended to insert the results for Asian dust and Arizona Test Dust into the figure. Even if they are not lying perfectly on the line, it illustrates better the consistency of your results. By the way, the information about the depolarization ratio of silicate and hematite is doubled (once next to the figure and once on the dashed line).*"

Thank you for your comment, we plotted the proposed correction as shown below. While the dust PDR should lie in between $\delta_{Sil}$ and $\delta_{Hmt}$, we observe that $\delta_{Sil}$ then becomes lower that $\delta_{Arizona}$. Since both Arizona and Asian dust are complex mixtures of more than two components (Sil, Hmt), their PDR may exceed that of pure components ($\delta_{Sil}$ and $\delta_{Hmt}$) as is the case here for Arizona. It is not surprising as the fraction of hematite is low in ATD, and so the potential effect of other components is more visible. As a result, the figure we present below may confuse future readers. To improve the readability of our contribution and avoid confusion, we then chose not to add $\delta_{Arizona}$ and $\delta_{Asian}$. To compare a mixing rule with laboratory evaluations, we should consider more components mixtures, which is beyond the scope of this study, as stated in our manuscript. Thank your for the proposed improvement on the repetitive information on the PDR, which we corrected.

[Figure]

"*Eq. 8: Indices should not be in italic.*"
Thank you for your comment, we did the correction.

"*The figures should be provided in higher resolution, with larger fonts. In their current state, they are very difficult to read.*"
Thank you for your comment, we did the correction.

"*Sections 4.3 and 5 are rather repetitive. I suggest merging those sections into one to avoid text repetitions.*"
Thank you for your comment. In line of our answer to Reviewer #1's comment on the same topic, we rather chose to rename the title of Section 5 to "Summary and conclusion". We expect it will help future readers.

---

## Author Comment (AC2)

Investigating the dependence of mineral dust depolarization on complex refractive index and size with a laboratory polarimeter at 180.0° lidar backscattering angle
Manuscript by A. Miffre, D. Cholleton, C. Noël and P. Rairoux
Submitted to Atmospheric Measurement Techniques (AMT).

We thank both reviewers for the time they spent to review our manuscript. The provided comments add value to our manuscript and may help future readers and we are grateful for that. For the sake of clarity, each comment is first recalled in italics, then our answer is given, together with a list of changes made to the manuscript, which also appear in red-lined version in the revised manuscript.

**Author's response to Reviewer #1 's comments**

*Reviewer #1: The paper presents laboratory measurements of the particle linear depolarization ratio (PLDR) for dust particle samples of different size and refractive indexes (RIs). The work is worth publishing in AMT journal, since it provides useful insights for the dust PLDR measurements with lidars in the ambient atmosphere.*

➢ We thank Reviewer #1 for the substantial amount of time she / he spent to review our manuscript. We are pleased to see that our work provides useful insights for the dust PLDR measurements with lidar and is then worth publishing in AMT journal.

*The use of English in the manuscript though is not optimum, and needs major re-right, focusing especially on the grammar and syntax used. Some corrections are provided here, but the authors are strongly advised to check the manuscript thoroughly, and improve it. Moreover, discussion on the dependence of PLDR on the dust shape and on larger dust particles (with diameter >20μm) is missing and should be included, at least in the form of discussion. Also, please be more specific about the definition of the QWP ψ angle. You use 2 or 3 different discerptions for this angle in the manuscript.*

➢ Thank you for your comments. For the sake of clarity, we answer point by point below and provide corrections that appear in red-lined version in the revised manuscript. Concerning the Ψ-angle definition, we agree that we could have been clearer though its definition is given in the caption of Figure 2. This angle modulates the polarization state of the laser by rotating the QWP.

**Major and general comments:**

*Reviewer #1, comment 1: The maximum size (diameter) considered in the study is 10μm (as shown in Fig. 1), excluding the full size range of dust particles in the atmosphere (e.g. Ryder et al., 2019). Include this info in the abstract, introduction and discussion.*

➢ **Answer to Reviewer's 1 comment 1:** We thank Reviewer #1 for her/ his comment which provides us with the reference to the paper by Ryder et al. (2019), which highlights the role of coarse and giant dust particles on the retrieved dust mass concentrations. In our experiment, according to the manufacturer, giant dust particles are present at a low number concentration. They represent 4.5 % of the volume size distribution. The maximal size (diameter) considered in our study is then more than 10 μm. As in (Järvinen et al., 2016), our size instruments could not however measure such large dust particle sizes, with diameter above 10 μm, since we do not have the required specific instrumentation to address these particles, such as the light shadowing technique operated by Ryder et al. (2019). As the number of these giant dust particles is low in our experiment, a nice outlook of our work will be to carry out a dedicated study on giant dust particles. According to Ryder et al. (2019), this will allow to specifically study fresh events (under

12h since uplift). We thank Reviewer #1 for her / his comment, which adds value to our manuscript.

➤ **Changes made to the manuscript:** As asked for by Reviewer #1, we modified our manuscript to consider reference to Ryder et al. (2019):
- In the Abstract, where we specified the considered size range, from 10 nm to more than 10 μm.
- In the Introduction, we specified the size range of our dust samples: "According to the manufacturer, the size distribution of our dust samples ranged from 10 nm to more than 10 μm in diameter."
- In Section 2.2, dedicated to the SD of our samples, we added: "As in Järvinen et al. (2016), our size instruments could not measure dust particles with diameter above 10 μm. According to the manufacturer, such giant particles (Ryder et al., 2019) are however present in our dust samples, at a low number concentration."
- In the Discussion section, we added the following sentence: "Also, it would be interesting to investigate giant dust particles (Ryder et al., 2019)."
- In the Outlooks section, we also modified our manuscript accordingly: "Giant dust particles, whose importance has been highlighted by Ryder et al. (2019), would likewise be interesting to study specifically. "

**Reviewer #1, comment 2:** *The dependence of PLDR on dust particle shape is not included in this analysis. Include this in the abstract, introduction, discussion and discuss in the manuscript.*

➤ **Answer to Reviewer's 1 comment 2:** Thank you for your comment. Though our π-polarimeter accounts for the irregular shape of mineral dust (see line 65), as in other existing laboratory light scattering experiments (see line 66 for references), the shape of our samples cannot be varied in a controlled manner, as we do for instance for the size distribution. We agree that the dependence of the PLDR with shape is an important issue. In the absence of possibility to vary it experimentally, we referred to the state-of-the-art literature. Light scattering numerical models (Gasteiger et al., 2011; Luo et al., 2022 ; Zubko et al., 2013) tackled this important issue as we quoted in the introduction. Our group itself contributed to study the influence of the dust particle shape on the lidar PDR (Mehri et al., 2018), as quoted at line 58. Here, we then focused on the PLDR dependence with size and complex refractive index. The shape dependence of PLDR becomes important when hematite is not involved: indeed, hematite exhibits a large imaginary part of complex refractive index, and according to Wiscombe (1986), "*when the imaginary index is large, nonsphericity can often be safely neglected*". This conclusion was also drawn by Mishchenko et al. (1997) who noted that the effect of particle shape on scattering properties of nonspherical particles become weaker with increasing the imaginary part of the refractive index. Likewise, Meland et al. (2011), who studied hematite at 470, 550 and 670 nm wavelengths, obtained different PLDR for different shape models only at wavelength 660 nm where hematite is less absorbing. They hence concluded that "*particle shape become important when the imaginary refractive index value is not large (even for a large real value)."* Nevertheless, the dependence of the PLDR on the dust shape remains important to study when hematite is less or not involved. Studying the shape dependence of the PLDR would then be a nice outlook of this work, however beyond the scope of this paper, which focuses on the dependence of the PLDR with size and complex refractive index, as emphasized in our title. We thank Reviewer #1 for this comment which adds value to our manuscript.

➤ **Changes made to the manuscript:** As asked for by Reviewer #1, we modified our manuscript as follows:
- In the Abstract: we modified the original sentence which now becomes: "As a result, when the strong light absorbent hematite is involved, the dust lidar PDR primarily depends on the particles complex refractive index and its variations with size and shape are less pronounced.

When hematite is less or not involved, the dust lidar PDR increases with increasing sizes, though the shape dependence may then also play a role. "

- In the Discussion section: we stressed the importance of the shape dependence of the PLDR by completing former line 370 (see below in italics) : "As a result, when the strong light absorbent hematite is involved, the dust lidar PDR primarily depends on the particles complex refractive index and its variations with size *and shape* are less pronounced. When hematite is less or not involved, the dust lidar PDR increases with increasing sizes*, though the shape dependence may then also play a role*." We likewise added: "The shape dependence of the hematite PDR is weak due to its large imaginary part of complex refractive index: following Wiscombe (1986) or Mishchenko et al. (1997), the effect of particle shape becomes weaker with increasing imaginary part of the refractive index, a conclusion also drawn by Meland et al. (2011)."

**Reviewer #1, comment 3:** *"PDR" should be changed to "PLDR" throughout the manuscript, since "PDR" may also denote to e.g. the particle circular depolarization ratio (PCDR). Moreover, the use of word "lidar" is not necessary, thus change "lidar PDR" to "PLDR". Do not use parentheses to provide values in the manuscript, unless necessary. For example, write 180°±0.2° instead of (180°±0.2°), or 355, 532nm instead of (355,532) nm. Change throughout the manuscript.*

➢ **Answer to Reviewer's 1 comment 3:** Thank you for your comment. We indeed deal with the linear particles depolarization ratio, as implied by its definition in Equations (3) and (4). At specific backscatter angle of 180.0°, the relationship between PLDR and PCDR is well-known anyway (Mishchenko et al., 2002). As our paper is mainly for lidar purposes (see line 10), we chose to note the PLDR simply as PDR, as done in the lidar community. Otherwise, the distinction between the lidar PDR and the PDR is stated in Eqs. (3) and (4), to emphasize the novelty of our work.

➢ **Changes made to the manuscript:** To account for Reviewer #1's comment, we added the word linear at line 186 and the following sentence to Section 3.2: "The PDR stated in Eq. (3) is the linear PDR, which can be related to the circular PDR if need be (Mishchenko et al., 2002)." We also did the corrections for the parentheses. We not changed PDR to PLDR all along the manuscript, thus following Reviewer #2's recommendation of changing it at least one instance.

**Specific comments:**

For clarity, we gathered the comments dealing with the same topics: i) on the dust particle size (in line of above comment 1), ii) on the dust particle shape (in line of above comment 2), iii) other scientific comments, iv) grammar and syntax propositions for which we thank Reviewer #1 for the proposed corrections. On this latter point, we considered only the comments which helped to clarify the manuscript to help future readers. All the corrections made can be followed in red-lined version.

**i)     Comments on the dust particles size (in line of above comment 1)**

*Line 80, "… the dust lidar PDR actually depends on the complex refractive index …": It also depends on the size and the shape of the dust particles, include this info here.*

At specific line 80, our concern is dedicated to the wavelength dependence of the dust lidar PLDR. For that reason, we focused on the complex refractive index dependence of the PLDR. We agree that the PLDR also depends on the dust particles size and shape, as we explained at line 50.

*Line 98, "… to particle sizes larger than 800 nm…": Provide the minimum and maximum diameters of dust particles in the samples used in this study. Include comment on the limitations of this study with respect to the size range, considering the absence of larger dust particles, with diameters >20μm, which are present in the ambient atmosphere, as measured in e.g. Ryder et al. (2019).*

See our answer to above Comment 1, where to account for the Reviewer 's comment, we modified the manuscript accordingly.

*Lines 144-145, "The coarser… regions,": Rephrase as "The coarser SD plotted with a grey line in Fig. 1. This SD is more representative of mineral dust particles close to dust regions, although it does not cover the full range of large dust particles measured close to dust sources, showing particles with diameters >50µm (e.g. Ryder et al., 2019)."*
We agree and did the correction.

*Lines 146-147: Measurements of long-range transported dust particles have shown particles with diameters >20µm (e.g. Weinzierl et al., 2017), thus the SD plotted with the black line is more representative of fine dust and not of long-range transported dust. Rephrase accordingly.*

In Weinzierl et al. (2017) is plotted in Figure 9 the number size distribution after transatlantic transport, which exhibits very few particles for sizes above 20 µm (less than $10^{-3}$ in dN/dLogD). Such a size distribution agrees with that of our samples, where, as explained in our answer to Comment 1, such giant particles are present at a low number concentration according to the manufacturer. To account for Reviewer's 1 comment, we added reference to Weinzerl et al. (2017). We thank you for this reference.

**ii)    Comments on the dust particle shape (in line with comment 2)**

*Lines 359-362, "To highlight… hematite lidar PDR.": Discuss the effect of size and shape of these two different samples.*
We checked the SD were the same. For the shape dependency, we refer to our answer to Comment 2. As hematite is involved, the shape effect is not pronounced, due to its large imaginary part of complex refractive index.

*Line 365: Discuss also the effect of shape.*
*Line 366: The works of Kahnert (2015) and Kahnert et al. (2020) do not consider the coarser dust particles, thus the effect of shape is expected to be lower. Please discuss.*
This is in line with our answer to above Comment 2, the shape effect is not pronounced for hematite as proven by the references we there quoted.

*Discussion Section: No discussion is provided on the effect of dust shape. Comment on this and highlight the lack of this analysis in the Section*
We refer the Reviewer to our answer to above Comment 2 where, to account for this Reviewer's comment, we modified the manuscript accordingly. For clarity, we recall that our manuscript is dedicated to the PLDR dependence with the size and the complex refractive index, as underscored by our title.

*Lines 403-404: Discuss shape dependence, as well.*
This is in line with our answer above Comment 2 where we modified the manuscript accordingly. At lines 403-404 however, we deal with the most general case, so we agree to add that the shape dependence may then play a role, depending on the hematite proportion. We modified our manuscript accordingly: "In the most general case, the dust lidar PDR hence appears as a complex function of the particles mineralogy, SD, wavelength and shape."

*Lines 415-417: Include also that the investigation of the dependence of PLDR on the dust particle shape is not included in this analysis.*
The sentence at lines 415-417 is dedicated to a given shape and not related to the dependence of the PLDR with the particles shape: "While mineral dust exhibits a complex and highly irregular shape, which

is difficult to model mathematically and numerically, our laboratory approach allows accounting for the real shape of mineral dust."

*Lines 433-437, "As a result… lidar instruments.": Provide comment on particle shape.*
See again our above answers to Comment 2.

**iii) Other scientific comments**

*Line 11, "…accurate values…": Quantify by providing the retrieval uncertainties of PLDR here.*
This information is given at lines 14 and 15 of the original manuscript. We added the information (< 1 %) to the abstract.

*Line 57, "… (Lindqvist et al., 2014) …": Include also the work of Gasteiger et al. (2011).*
Thank you for your comment. We quoted the paper by Gasteiger et al. (2011) in our previous papers (see Mehri et al., 2018). We hence agree and did the correction.

*Line 59, "… (Luo et al., 2022) …": Include also the work of Huang et al. (2022).*
We agree and added this reference. This was not however feasible when submitting our manuscript last June for the manuscript by Huang et al. (2022) started its AMT-discussion stage last 25th of October.

*Line 84, "… accurate values…": Provide uncertainty of the retrieved PLDR to quantify "accurate".*
The accuracy is below 1 % as stated in the Results. We added this information (< 1 %) at line 84.

*Line 124, "… m=2.25+0.9i…": Contact Go et al. to verify this value.*
Go et al. used the results published in Scanza et al., ACP, 2015 where m = 2.13 + 0.94i at 355 nm wavelength (3.07 + 0.55i at 532 nm wavelength). Thank you for your comment. The imaginary part of complex refractive index is the same, which is the key point of our study. To account for Reviewer #1's comment, we modified our manuscript accordingly: "the real and imaginary part of the hematite CRI were reviewed by Scanza et al. (2015): m = 2.13 + 0.94i at 355 nm wavelength (3.07 + 0.55i at 532 nm wavelength)."

*Lines 129-131, "Effective… wavelength.": Rephrase as "Miffre et al. (2016) derived the values of m=…. at 355 nm and m=… at 532 nm, using effective-medium approximations.*
It seems to us that this rephrasing adds no new information on the involved physics. Hence, after carefully checking the English of our sentence, we chose to keep our own writing of the sentence.

*Line 135: Provide the CRI of Asian dust.*
Thank you for your comment. The CRI of Asian dust, evaluated from effective medium approximation, is m = 1.70 + 0.09i at 355 nm wavelength (1.72 + 0.03i at 532 nm). We added these values in the manuscript.

*Lines 194-195, "The deviation… since…": Replace with "The deviation of $F_{11,\lambda}$, $F_{12,\lambda}$ and $F_{22,\lambda}$ at near-backscattering angles, compared to their value at exact backscattering angle cannot be easily evaluated with scattering calculations, since…". Include here also numerical approximations used to calculate the scattering properties of irregularly-shaped dust particles for lidar applications, e.g. the work of Gasteiger et al. (2011), Konoshonkin et al. (2020).*
At lines 194-195, we stress that no analytical light scattering theory exists for complex-shaped mineral dust allowing to evaluate this deviation. J. Gasteiger himself underscores in his more recent work (Gasteiger et al. 2018) on light-scattering calculations the difficulty to model the depolarization ratio of large particles at backscattering angle (see Supplementary material of his paper), when the shape is

complex, so our laboratory polarimeter may help, as it accounts for the real complex shape of mineral dust. Konoshonkin et al. (2020) focuses on ice crystals, which are not our concern, and we already quoted Gasteiger et al. (2011).

*Line 206, "In Miffre et al. (2016), for the first time to our knowledge, …": Miffre et al. (2016) refer that the π-polarimeter was the one built in David et al. (2013). Provide info and the corresponding reference here.* Thank you for reading our former paper also. If we introduced the basic principle in David et al. (2013), it is not necessary to quote again this reference which is included in Miffre et al. (2016) which is then more complete.

*Line 221, "… the lidar PDR … (Mishchenko et al., 2002).": Replace with "… the PLDR at 355 and 532 nm simultaneously, for an aerosol sample.". I do not understand why you use the work of Mishchenko et al., (2002) as a reference here.* The work by Mishchenko et al. (2002) is here quoted as the Pi-polarimeter relies on the scattering matrix formalism, extensively described in this reference textbook.

*Line 246, "Accurate evaluations of the dust PDR…": Provide uncertainties, otherwise replace with "The dust PLDR…".* The accuracy is that given in the Results section in Tables 1 and 2. It is below 1 % and even lower for most samples. As it is complicated to detail that point at line 246, we finally chose to remove the word accurate, to invite the readers to precisely read Tables 1 and 2.

*Lines 255-258, "The curve … size-dependent.": $F_{11}$- $F_{22}$ and $F_{12}$ are both shape, size and RI depended. Discuss and change text accordingly.* $F_{11}$-$F_{22}$ is a commonly used metrics in lidar for discussing on the shape since it is null for spherical particles. For $F_{22}$, to account for the Reviewer's comment, we added "though it is also shape dependent".

*Line 264, "Accuracy on the retrieved…": Replace with "Accuracy of the retrieved…". Include in this Section a reference of the work of David et al. (2013) since more info about what is discussed, is provided there.* In David et al. (2013), a single wavelength was used, without dust particles, and the fluctuations in the number concentration were not circumvent. The present contribution fills this gap. Our paper David et al. (2013) then contains less information.

*Line 277, "…any possible mismatch…": What do you mean by this? Please rephrase/explain.* By mismatch, we intend the potential uncertainties that may affect the Pi-polarimeter with respect to the scattering plane.

*Line 290: Provide the full equation for $I_\lambda(\vartheta_o)$, including the Mueller matrix sequence, similar to Eq.5.* The full equation is $I_\lambda(\theta_0) = [1, 0, 0, 0]^T [\mathbf{F}_\lambda(\theta_0)][\mathbf{QWP}(\psi)][\mathbf{PBC}][1, 1, 0, 0]^T$ , which we added to the revised manuscript.

*Line 291-293, "One the variations…number concentration". Provide example (with plots) here or in the appendix.* We here provide example with plots to show this intermediate step in data processing. By accounting for the rapid fluctuations in the particle number concentration, the statistical uncertainty on the backscattered light intensity is reduced.

[Figure]

*Line 296, "…calculated by considering the covariance of $I_\lambda$ and $I_\lambda(\vartheta_o)$.": Provide more info about this calculation.*

Since $I_{\lambda,N} = I_\lambda/I_\lambda(\theta_0)$, the covariance $\sigma_{I_\lambda I_\lambda(\theta_0)}$ contributes to the uncertainty on $I_{\lambda,N}$ at a rate $-2I_\lambda \sigma_{I_\lambda I_\lambda(\theta_0)}/I_\lambda^3(\theta_0)$. To account for Reviewer #1's comment, we added this sentence to our manuscript in Section 3.5.

*Line 298, "… standard deviations…": Provide more info: is this the variability of the measurements? Is this the measurement error? Or a combination?*

The standard deviation is calculated as explained above and mostly dominated by statistical uncertainties since our biases are minimized, as explained in Section 3.5. We added this information to the caption of Table 1: "The uncertainty on $F_{22,\lambda}/F_{11,\lambda}$ is mostly dominated by statistical uncertainties since our biases are minimized, as explained in Section 3.5."

*Line 309-310, "The uncertainty of $F_{22,\lambda}/F_{11,\lambda}$ … π-polarimeter.": Provide the methodology for deriving the uncertainties.*

As explained at lines 245-246, the uncertainty on $F_{22,\lambda}/F_{11,\lambda}$ originates from the evaluation of $b_\lambda/a_\lambda$, itself deduced from the least-square fit adjustment of the normalized backscattered light intensity. We added this information to the caption of Table 1: "The uncertainty on $F_{22,\lambda}/F_{11,\lambda}$ is deduced from the evaluation of $b_\lambda/a_\lambda$, itself deduced from the least-square fit adjustment of $I_{\lambda,N}$ ".

*Line 319, "Normalized backscattered light intensity": Explain with what you normalized the backscattered light intensity.*

This information is given at lines 294-295 of the original manuscript: "*As a result, in the light backscattering curves presented in Section 4, the plotted quantity is the normalized backscattered light intensity $I_{\lambda,N} = I_\lambda/I_\lambda(\theta_0)$*".

*Line 410: Replace "Conclusions" with "Summary and conclusions", since you also provide a summary of the work here.*

Thank you for your comment, we agree and did the modification in the manuscript.

*Line 419, "… accurate…": Quantify (or delete).*

See answer to above comments: the answer is given in the corresponding Tables 1 and 2. It is below 1 %.

*Line 245-246, "… then adjusting … then $b_\lambda/a_\lambda$.": Provide here the methodology you use to derive $b_\lambda/a_\lambda$ from the measurements of $I_\lambda(\psi)$. Is it a least-squares fit? Discuss the uncertainties of the retrieved $b_\lambda/a_\lambda$.*

Yes, a least-squares fit is applied and the standard errors are propagated by following Equation (7) to evaluate the uncertainties on the retrieved $b_\lambda/a_\lambda$.